



# Validation of Aeolus winds using radiosonde observations and NWP model equivalents

Anne Martin[1], Martin Weissmann[2], Oliver Reitebuch[3], Michael Rennie[4], Alexander Geiß[1], and Alexander Cress[5]

[1]Ludwig-Maximilians-Universität, Meteorologisches Institut, München, Germany
[2]Universität Wien, Institut für Meteorologie und Geophysik, Wien, Austria
[3]Deutsches Zentrum für Luft- und Raumfahrt e.V. (DLR), Institut für Physik der Atmosphäre, Oberpfaffenhofen, Germany
[4]European Centre for Medium-Range Weather Forecasts (ECMWF), Reading, UK
[5]Deutscher Wetterdienst (DWD), Offenbach am Main, Germany

**Correspondence:** Anne Martin (anne.martin@physik.uni-muenchen.de)

**Abstract.** In August 2018, the first Doppler Wind Lidar, developed by the European Space Agency (ESA), was launched on board the Aeolus satellite into space. Providing atmospheric wind profiles on a global basis, the Earth Explorer mission is expected to demonstrate improvements in the quality of numerical weather prediction (NWP). For the use of Aeolus observations in NWP data assimilation, a detailed characterization of the quality and the minimization of systematic errors is crucial.

This study performs a statistical validation of Aeolus observations, using collocated radiosonde measurements and NWP forecast equivalents from two different global models, the ICOsahedral Nonhydrostatic model (ICON) of Deutscher Wetterdienst (DWD) and the European Centre for Medium-Range Weather Forecast (ECMWF) Integrated Forecast System (IFS) model, as reference data. For the time period from the satellite's launch to the end of December 2019, comparisons for the northern hemisphere $(23.5 - 65°N)$ show strong variations of the Aeolus winds bias and differences between the ascending and descending

orbit phase. The mean absolute bias for the selected validation area is found to be in the range of 1.8 - 2.3 m s$^{-1}$ (Rayleigh) and 1.3 - 1.9 m s$^{-1}$ (Mie), showing good agreement between the independent reference data sets. Due to lower representativeness, the random differences are larger for the validation using radiosonde observations compared to the model equivalent statistics. To achieve an estimate for the Aeolus instrumental error, the representativeness errors for the comparisons are determined, besides the estimation of the model and radiosonde observational error. The resulting Aeolus errors estimates are in the range

of 4.1 - 4.4 m s$^{-1}$ (Rayleigh) and 1.9 - 3.0 m s$^{-1}$ (Mie). Investigations of the Rayleigh wind bias on a global scale show that in addition to the satellite flight direction and seasonal differences, the systematic differences depend on latitude. A latitude based bias correction approach is able to reduce the bias, but a residual bias of 0.4 - 0.6 m s$^{-1}$ with a temporal trend remains. Taking additional longitudinal differences into account, the bias can be reduced further by almost 50 %. Longitudinal variations are suggested to be linked to land-sea distribution and tropical convection that influences the thermal emission of the earth. Since

20 April 2020 a bias correction scheme has been applied operationally in the L2B processor, developed by the Aeolus Data Innovation and Science Cluster (DISC).





# 1  Introduction

Aeolus is a European Space Agency (ESA) Earth Explorer mission, launched on 22 August 2018 as part of the Living Planet Programme. With an estimated lifetime of three years, it is expected to pave the way for future operational meteorological satellites dedicated to observing the Earth's wind fields (ESA, 2008). Aeolus is a polar orbiting satellite, flying in a sun-synchronous dawn-dusk orbit at about 320 km altitude. Within seven days, the satellite covers nearly the whole globe. Aeolus carries only one large instrument, a Doppler wind lidar called ALADIN (Atmospheric LAser Doppler INstrument) which is the first European lidar and the first ever Doppler Wind Lidar (DWL) in space (Stoffelen et al., 2005; Reitebuch, 2012; ESA, 2008). ALADIN provides profiles of the line-of-sight (LOS) wind component perpendicular to the satellite velocity at an angle of $35°$ off-nadir from the ground up to 30 km.

The Aeolus mission primarily aims to demonstrate improvements in atmospheric wind analyses for the benefit of numerical weather prediction (NWP) and climate studies (Stoffelen et al., 2005; Rennie and Isaksen, 2019). Despite the advancement of the Global Observing System (GOS), there are still major deficiencies, the lack of accuracy are significant limitations of currently used wind observation methods (Källen, 2018). Accurate vertical profiles of the wind field from radiosondes, wind profilers, and commercial aircraft ascents and descents are mainly concentrated over continents in the northern hemisphere, whereas only a few profiles are available over the oceans and on most parts of the southern hemisphere. Atmospheric motion vectors derived from tracking cloud and water vapor structures in consecutive satellite images provide single-level winds with nearly global coverage but exhibit significant systematic and correlated errors due to uncertainties of their height assignment (e.g. Folger and Weissmann, 2014; Bormann et al., 2003). The vast majority of global observations consist of satellite radiances, mainly providing information on the atmospheric mass field (temperature, humidity, other trace gases, and hydrometeors). Wind information can only be retrieved indirectly from these observations, which is a particularly strong restriction in the tropics in the absence of geostrophic balance. The actively sensed globally distributed lidar LOS winds are therefore filling a major gap of the GOS, especially in the upper troposphere and the lower stratosphere, in the tropics and over the oceans (Baker et al., 2014; ESA, 2008). It has been shown that improvements are to be expected for short range forecasts of severe weather situations, the analysis of tropical dynamics, and for a better definition of smaller scale circulation systems in midlatitudes (e.g. Marseille et al., 2008; Tan and Andersson, 2005; Weissmann and Cardinali, 2007; Weissmann et al., 2012; Zagar, 2004). A crucial prerequisite for the use of meteorological observations in NWP data assimilation systems is a good knowledge of their statistical errors and the minimization of systematic observation errors. For this purpose, uncertainty assessment and validation through analytical comparisons with reference data is an essential requirement to obtain best benefits for NWP.

The Aeolus direct detection wind lidar (ALADIN) is operating in the ultraviolet spectral region (354.8 nm). The laser emits pulses of about 60 mJ at a frequency of 50.5 Hz. A Cassegrain telescope with a diameter of 1.5 m collects the backscattered signal, which Doppler shift is analyzed by a dual channel receiver to measure backscattered signals from both, molecules (Rayleigh channel) and particles (Mie channel) (ESA, 2008; Reitebuch, 2012). This complementarity of the two channels allows for broad vertical and horizontal data coverage in the troposphere. In preparation of the Aeolus mission, a prototype of the satellite instrument, the ALADIN Airborne Demonstrator (A2D), was deployed to test the wind measurement principles





under real atmospheric conditions in several measurement campaigns, and to provide information on quality control algorithms (Lux et al., 2018). Two airborne validation campaigns with operation base at DLR (Deutsches Zentrum für Luft- und Raumfahrt e.V.) Oberpfaffenhofen were already performed within the first ten months after the satellite's launch. Deploying the A2D and a 2-$\mu$m DWL as reference, wind data for the first experimental comparisons with the Aeolus wind product and model wind data from the European Centre for Medium-Range Weather Forecasts (ECMWF) were provided. Detailed information and results have been published in Lux et al. (2020) and Witschas et al. (2020). A further comparison of Aeolus winds with a direct-detection Doppler lidar (the LIOvent instrument at the Observatoire de Haute-Provence) for a time period at the beginning of 2019 is discussed in Khaykin et al. (2020). As part of the German initiative EVAA (Experimental Validation and Assimilation of Aeolus observations), this paper presents the evaluation of Aeolus winds using operational collocated radiosonde data from the GOS as reference. Besides, observation monitoring statistics from the global ICOsahedral Nonhydrostatic model (ICON) of Deutscher Wetterdienst (DWD) and the ECMWF Integrated Forecast System (IFS) model are analyzed to corroborate the results and investigate dependencies and possibles causes of systematic deviations. The text is structured as follows. First, an overview and description of the data sets used for the evaluation of the Aeolus winds is provided. Collocation criteria are specified and the statistical methods for the comparison are described. Section 3 presents a time series of the validation, focusing on the temporal evolution of systematic and random differences between the Aeolus observations and the reference data sets. To derive error estimates for the Aeolus winds, the representativeness errors of the comparisons are estimated using analysis data from the regional model COSMO (Consortium for Small-scale MOdeling) of DWD and high resolution ICON Large Eddy Model (LEM) data. In Section 4, the Aeolus Rayleigh channel bias is investigated in more detail and two bias correction approaches are evaluated. Finally, the results are discussed and summarized.

## 2 Data and method

The Aeolus Level 2B (L2B) product is evaluated using collocated radiosonde observations of the GOS and short-term model forecast equivalents (first guess departure statistics) of the global model ICON of DWD and the ECMWF model as reference.

### 2.1 Aeolus L2B wind product

The Aeolus L2B product contains the Horizontal LOS wind component (HLOS) observations suitable for NWP data assimilation (ECMWF: Rennie et al., 2020). The majority of wind data are provided by the Rayleigh channel. In clear conditions, the Rayleigh wind coverage is from the surface up to 30 km. The Mie signals are strong within optically thin clouds, on top of optically thick clouds, and cover the atmospheric boundary layer as well as aerosol layers for clear sky conditions. Each Aeolus measurement is an accumulation of 20 laser pulses which corresponds to a horizontal resolution of about 2.9 km. To achieve a sufficient signal to noise ratio to comply with the stringent wind accuracy requirements, observations are processed by averaging up to 30 individual measurements. The resulting HLOS wind is therefore a horizontal average over 86.4 km. For the Mie channel, the horizontal integration length of the wind measurements was decreased to approximately 10 km after 5 March 2019, taking benefit of the higher signal to noise ratio of cloud returns (Matic et al., 2019). In addition to the HLOS



observations, the Aeolus L2B processor developed by ECMWF and the Royal Netherlands Meteorological Institute (KNMI) provides an observation instrument noise estimate. Furthermore, to avoid systematic errors, corrections for the temperature and

pressure dependence of the Rayleigh winds are performed using a priori information from the ECMWF model interpolated along the Aeolus track (Dabas et al., 2008). The measurements within an observation are classified into an observation type, clear or cloudy, using measurement-scale (2.9 km) optical properties, such as scattering ratio. Wind retrievals are performed for both channels and observation types. The vertical resolution varies from 0.25 km near surface to 2 km in the highest bins, with a total of 24 bins. The processing at ECMWF is performed in near-real-time, thus measurements are delivered within

three hours. More detailed information about the L2B processor retrieval algorithm can be found in ECMWF: Rennie et al. (2020). As Aeolus is a novel mission, the processing algorithms have been evolving since launch. Different processor baselines (in this study 2B02 - 2B07) and various updates led to different observation quality in different time periods. A consistent reprocessed data set with unique processor settings is not available yet. Furthermore, the instrument performance varied over the time period assessed in this study, which includes the missions Commissioning Phase (CP) from launch until the end of

January 2019, the late Flight Model A (FM-A) laser period until mid of June 2019, and the FM-B laser period until the end of December 2019. Information about the actual performance of the Aeolus wind lidar and a discussion of the systematic and random error sources can be found in Reitebuch et al. (2020) and Rennie and Isaksen (2020). For the validation, only valid Rayleigh clear and Mie cloudy winds (from now on referred to as Rayleigh and Mie) between 800 and 80 hPa are used. A distinction is made between the ascending orbital pass, when the satellite moves north, and the descending orbital pass when

the satellite moves south. Based on a compromise between the quality of the data set and the number of observations that pass the quality control, Rayleigh winds with an estimated error greater than 6 m s$^{-1}$ and Mie winds with an estimated error greater than 4 m s$^{-1}$ are excluded for the evaluation.

## 2.2 Radiosonde data and collocation criteria

Radiosonde observations generally provide very accurate information on the true wind conditions. Given that radiosonde wind

data are direct in situ measurements, the inherent errors (e.g. instrument errors) are small compared to errors of satellite-based instruments. That makes them well suited to serve as reference data set for the true atmospheric state for the validation of Aeolus HLOS winds. Furthermore, the observation errors can be assumed to be uncorrelated between different radiosondes. At ECMWF, radiosonde feedback files are created from the Observational DataBase (ODB) at the end of the IFS analysis and archived in the Meteorological Archival and Retrieval System (MARS). For stations where ECMWF is assimilating BUFR

(Binary Universal Format for the Representation of meteorological data) data, the balloon drift is taken into account by splitting into groups of data from each 15 min time slot. Radiosonde feedback files from alphanumeric reports only contains the time and position of the radiosonde's launch, but not the time and position of the individual wind observations. Due to the radiosonde drifts during the ascent and the ascent time additional errors arise. Seidel et al. (2011) evaluated characteristic values of average drift distances to be 5 km in the mid troposphere, 20 km in the upper troposphere, and 50 km in the lower stratosphere, tending

to be larger in midlatitudes than in the tropics. A few individual radiosondes are found to drift up to 200 km. Estimates of the ascent time of the balloon range from 5 min, when it reaches 850 hPa, up to 1.7 h at 10 hPa. These values should be taken





into account when defining collocation criteria for comparisons with radiosondes. In this study, all radiosonde observations that are within 120 km horizontal, 90 min temporal, and 500 m vertical distance from the Aeolus measurements are used for the validation statistics. For each location, the radiosonde HLOS wind component is computed as linear function of the zonal

wind component $u$ and the meridional wind component $v$ as

$$\text{HLOS} = -u \cdot \sin(\phi) - v \cdot \cos(\phi), \tag{1}$$

where $\phi$ is the L2B azimuth angle, which is defined clockwise from north of the horizontal projection of the target to satellite pointing vector. Since radiosonde observations are rare in the southern hemisphere and polar regions, the analysis concentrates on the midlatitudes of the northern hemisphere ($23.5 - 65°$N). To achieve a sufficiently large data set, statistics for one day are

based on a running mean over seven days.

### 2.3     Model data for the validation

For a more comprehensive global assessment, the validation results of Aeolus winds with radiosondes are supplemented by a comparison to model equivalents from the global model ICON of DWD and the ECMWF IFS model. Due to the inhomogeneous spatial and temporal distribution of radiosondes, the model data continue to serve as a basis for further investigations

of longitudinal and latitudinal bias dependencies. The global NWP system of DWD combines a three-dimensional variational technique (3D-Var) with a Local Ensemble Transform Kalman Filter (LETKF) to produce consistent initial states for an ensemble forecasting system using the ICON model. The deterministic first guess forecast is used to calculate the observation first guess departures (O-B). In contrast to the ECMWF data assimilation system (4D-Var), the observations are not used at their actual time, but all observations within an observation window ($\pm$ 1.5 hr around the analysis time) are assumed to be

valid at the analysis time. The Aeolus observational feedback files of the ECMWF IFS model, as well as the monitoring files of the ICON model used for this study, include all observations that were screened by the data assimilation system, but not influencing the analysis. At ECMWF, the Aeolus HLOS winds are assimilated operational since 09 January 2020, at DWD the operational assimilation started on 19 May 2020.

To ensure comparable data sets for the radiosonde and the ECMWF and DWD model validation of the Aeolus winds, only the

nearest O-B value per radiosonde collocation is used for the model validation statistics. To put the regional validation results in a global context, additionally a global statistic with the ECMWF O-B values is calculated, using a similar approach for limited regions, which are chosen to be $10°$ latitude x $10°$ longitude, and limited periods of seven days.

### 2.4     Model data for the estimation of the representativeness error

To evaluate representativeness errors for the Aeolus wind validation, analyses data of the regional COSMO-DE model of five

seven-day periods (February, April, June, October, and December 2016) are used. The COSMO-DE model covers Germany, Switzerland, Austria, and parts of other neighboring states and has a horizontal grid spacing of 2.8 km and 50 levels in the vertical. To determine the effect of unresolved scales in the COSMO-DE analyses, the results are compared to a three day (3 to 6 June 2016) large-eddy simulation with the ICON model centered over Germany with 150 m horizontal resolution and 150





levels in the vertical. The data are only used up to 12 km to avoid influences of large model errors and uncertainties of the
simulation in the stratosphere.

## 2.5   Statistical characteristics

The following outlines the applied statistical metrics. Using the forecast of NWP models as reference, the bias estimate is
described by the mean first guess departure:

$$\text{BIAS} = \frac{1}{N} \sum_{i=1}^{N} (y_i - H(x_{\mathbf{b},i})), \tag{2}$$

where $y$ is the Aeolus HLOS wind observation, $x_{\mathbf{b}}$ the state vector of the short-term model forecast (background) and $H(.)$ the
observation operator. Given that the model bias for long validation periods and large scales is usually small in comparison to
that of Aeolus observations, the mean difference between the Aeolus observations and the reference data can be referred to as
bias. In certain conditions, such as in jet stream regions, the tropical upper troposphere and the stratosphere, however, Aeolus
HLOS bias estimates based on NWP monitoring statistics should be treated with caution (Rennie, 2016).
The bias using the radiosonde measurements as reference is calculated according to

$$\text{BIAS} = \frac{1}{N} \sum_{i=1}^{N} (\text{HLOS}_{\text{L2B},i} - \text{HLOS}_{\text{Radiosonde},i}). \tag{3}$$

For quantifying random deviations, the standard deviation

$$\sigma = \sqrt{\frac{1}{N\text{-}1} \sum_{i=1}^{N} ((\text{HLOS}_{\text{L2B},i} - \text{HLOS}_{\text{reference},i}) - \text{BIAS})^2}, \tag{4}$$

as well as the scaled median absolute deviation (MAD)

$$\text{scaled MAD} = 1.4826 * \text{median}[|(\text{HLOS}_{\text{L2B}} - \text{HLOS}_{\text{reference}}) - \text{median}(\text{HLOS}_{\text{L2B}} - \text{HLOS}_{\text{reference}})|], \tag{5}$$

is determined. The MAD is a very robust measure for the variability of the Aeolus HLOS winds, being more resilient to single
outliers compared to the standard deviation. In case of a normally distributed data set, the MAD value multiplied by 1.4826
(scaled MAD) is identical to the standard deviation (Ruppert and Matteson, 2015).

## 3   Validation results - time series of comparisons with radiosonde observations, ICON model and ECMWF IFS model
equivalents

### 3.1   Systematic and random differences

For the time period from the first available L2B data after the satellite's launch up to January 2020, systematic and random
differences between the Aeolus HLOS winds, radiosondes and model fields are calculated. Figure 1 compares validation





results for the latitudinal band $23.5 - 65°$N using collocated radiosonde observations (blue) and O-B statistics of the global

NWP models of ECMWF (orange) and DWD (green), separated for Rayleigh clear and Mie cloudy winds, and for the two orbit

phases of the satellite. With the defined quality control and collocation criteria about 4500 Aeolus Rayleigh wind observations

and about 2300 Mie wind observations per seven day period are used for the validation statistics. Table 1 provides an overview

of the mean absolute differences and the mean scaled MAD of the whole period for the areas around the radiosonde locations

on the northern hemisphere and for a global statistic using the ECMWF (O-B) values. The Rayleigh winds mean absolute bias

using radiosonde observations and O-B statistics of the ICON and the ECMWF IFS model differ only in a range of about 0.40

m s$^{-1}$, the Mie winds mean absolute bias differs in a range of about 0.64 m s$^{-1}$. The smallest mean absolute bias estimate is

found for the ECMWF first guess departures. The mean scaled MAD constantly shows larger values for the validation using the

radiosonde data as reference compared to the model O-B statistics. This can be explained by a lower representativeness of the

radiosonde in situ observations. Besides the much smaller resolution of a radiosonde observation compared to the resolution

of a global NWP model, representativeness errors arise from the chosen collocation criteria and the spatial and temporal

displacement during the radiosonde ascents. Theses error sources are considered in the Aeolus HLOS winds error estimation

in the following Subsection 3.2. Comparing the two NWP models, the mean scaled MAD calculated with the ECMWF model

is on average about 0.14 m s$^{-1}$ smaller than when using O-B statistics of the DWD global model. Likely, this is mainly the

result of neglecting the temporal evolution within the assimilation window in the DWD system. The globally derived Rayleigh

winds mean absolute bias estimates, which are based on ECMWF first guess departures of limited areas ($10°$ latitude x $10°$

longitude) and periods (seven days) are slightly smaller compared to the model validation results of the restricted areas on the

northern hemisphere. For the Mie winds, the global statistic shows values in the range of the validation statistics around the

radiosonde collocations.

   Assessing the temporal development of the Aeolus wind bias, it is apparent that the quality of the observations varies over

time. To some extent, this is caused by six different processor baselines, and several updates of the calibrations files during the

selected time period, which makes the data partly inconsistent and incompatible. Right after the Aeolus launch, the Rayleigh

winds ascending phase exhibits a negative bias, whereas the descending phase is positive biased. With time, the Rayleigh

bias increases for both orbits. In January 2019, there was a reboot anomaly on the GPS unit on the satellite which led to the

ALADIN instrument being in a standby modus for around one month (grey shaded area). Right after the standby period, the

Rayleigh ascending bias reaches its maximum. For the descending orbit, the maximum occurs later in April 2019. The Mie

winds mean differences also show a positive trend within the first eight months, but the values are smaller compared to the

Rayleigh bias. The higher fluctuations in bias compared to the Rayleigh winds might be linked to the sparser coverage of the

Mie winds and the higher variability and larger model error when clouds are present. Related to an update of the processor

setting file in the end of May (Rennie and Isaksen, 2020), the estimated bias shows a sharp decline for both channels and orbit

phases. For the Rayleigh winds, the decrease is about 4 to 5 m s$^{-1}$, resulting in a negative bias, while the Mie winds bias

fluctuates around zero. Due to the decrease in the FM-A laser UV output energy, ESA switched to the second flight laser (FM-

B) in June 2019. Therefore, a second period without data occurs between 16 and 28 June 2019. The validation study continues

on 1 August 2019, when the new FM-B calibration files have been implemented. After the laser switch, the Mie winds bias



**Figure 1.** Time series from September 2018 to end of December 2019 of the bias, standard deviation and scaled MAD of Aeolus HLOS winds for the northern hemisphere ($23.5 - 65°$N), using collocated radiosonde observations (blue) and model equivalent statistics (O-B) around the collocation points of the ECMWF IFS model (orange) and the ICON model of DWD (green). **(a)**: Rayleigh clear winds, ascending orbit phase; **(b)**: Rayleigh channel descending orbit phase; **(c)**: Mie cloudy winds, ascending orbit phase; **(d)**: Mie channel, descending orbit phase. The background colors indicate the different processor baselines.

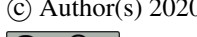

**Table 1.** Overview of the Aeolus HLOS winds mean absolute bias and mean scaled MAD from September 2018 to December 2019 for the northern hemisphere ($23.5 - 65°$N), restricted to the radiosonde collocation areas (3 top-most rows), and for a global statistic using the ECMWF model (bottom row).

| | Rayleigh ascending | | Rayleigh descending | |
| --- | --- | --- | --- | --- |
| | \| BIAS \| | 1.4826*MAD | \| BIAS \| | 1.4826*MAD |
| Radiosondes | 1.98 m s$^{-1}$ | 5.07 m s$^{-1}$ | 2.26 m s$^{-1}$ | 4.95 m s$^{-1}$ |
| ECMWF | 1.75 m s$^{-1}$ | 4.18 m s$^{-1}$ | 1.86 m s$^{-1}$ | 4.18 m s$^{-1}$ |
| DWD (ICON) | 1.84 m s$^{-1}$ | 4.25 m s$^{-1}$ | 2.07 m s$^{-1}$ | 4.27 m s$^{-1}$ |
| ECMWF global | 1.40 m s$^{-1}$ | 4.33 m s$^{-1}$ | 1.63 m s$^{-1}$ | 4.25 m s$^{-1}$ |

| | Mie ascending | | Mie descending | |
| --- | --- | --- | --- | --- |
| | \| BIAS \| | 1.4826*MAD | \| BIAS \| | 1.4826*MAD |
| Radiosondes | 1.41 m s$^{-1}$ | 4.00 m s$^{-1}$ | 1.90 m s$^{-1}$ | 3.92 m s$^{-1}$ |
| ECMWF | 1.31 m s$^{-1}$ | 2.70 m s$^{-1}$ | 1.26 m s$^{-1}$ | 2.79 m s$^{-1}$ |
| DWD (ICON) | 1.58 m s$^{-1}$ | 2.86 m s$^{-1}$ | 1.88 m s$^{-1}$ | 3.04 m s$^{-1}$ |
| ECMWF global | 1.45 m s$^{-1}$ | 2.53 m s$^{-1}$ | 1.55 m s$^{-1}$ | 2.52 m s$^{-1}$ |

fluctuations are reduced. The mean differences show quite constant and very small values for the late summer and autumn
months. The Rayleigh winds of the descending orbital phase exhibit a positive bias between 2 and 3 m s$^{-1}$ in August 2019,
tending to negative during the respective processor baseline period. The Rayleigh ascending wind bias varies between -3 and 0
m s$^{-1}$. Towards the end of the year 2019, when the Rayleigh bias is negative for both orbit phases, a sharp increase occurs in
mid-December. This is caused by a L2B processor bias correction update. The Mie winds mean differences are only slightly
increasing. All three independent reference data show very good agreement for the bias estimation, raising confidence that
the results are not determined by model biases. Besides the temporal changes in Aeolus Rayleigh and Mie winds quality, the
discrepancies between the ascending and descending orbit, mainly for the Rayleigh channel are a challenging issue for using
these data in NWP models. Significant differences occur especially in late summer and autumn. Assessing the mean absolute
values, the bias is larger for the descending than for the ascending orbit for both channels. For a more detailed analysis of the
Rayleigh bias, see Section 4.
The Rayleigh winds random difference calculated based on model O-B statistics varies between 3 and 6 m s$^{-1}$ within the
considered validation period. For the comparison with radiosonde observations, the mean random difference ranges from 4 up
to 7 m s$^{-1}$. The Mie winds random difference show in total smaller values, bu stronger fluctuations. Overall, a slight increase
in standard deviation and scaled MAD till summer 2019 is visible. This is likely associated with the energy decrease of the
FM-A laser over time. The laser switch led to reduced random differences for the Rayleigh channel. The Mie winds random
differences do not exhibit such clear changes, because the Mie return signal not only depends on the laser energy but also





on the presence of aerosols or hydrometeors. Since mid-October 2019, the Rayleigh winds random differences again show a small increase. Comparing the standard deviation and scaled MAD, no striking differences appear. On average, the standard deviation is about 0.20 m s$^{-1}$ larger than the scaled MAD, implying a few outliers in the statistics. To derive error estimates of the Aeolus HLOS winds, also the representativeness errors of the comparisons and errors resulting from the radiosonde

measurements and the NWP models have to be taken into account (see Subsection 3.2).

### 3.2 Estimation of the Aeolus HLOS wind error

The total variance of the differences between radiosonde observations and Aeolus HLOS winds $\sigma_{val}^2$ (squared scaled MAD) is the sum of the variance resulting from the Aeolus winds instrumental error $\sigma_{Aeolus}^2$, the variance resulting from the radiosondes winds observational error $\sigma_{RS}^2$, and the variance caused by the representativeness error $\sigma_{r\_RS}^2$ (Weissmann et al., 2005) (see

Equation 6a). For the comparison with model equivalents, the model representativeness error $\sigma_{r\_model}^2$ is used and $\sigma_{RS}^2$ is replaced by the model error $\sigma_b^2$ (see Equation 6b).

$$\sigma_{Aeolus} = \sqrt{\sigma_{val\_RS}^2 - \sigma_{r\_RS}^2 - \sigma_{RS}^2} \tag{6a}$$

$$\sigma_{Aeolus} = \sqrt{\sigma_{val\_model}^2 - \sigma_{r\_model}^2 - \sigma_b^2} \tag{6b}$$

As no model error estimate is available in the monitoring files of the ICON model, the Aeolus HLOS winds error is only

assessed for the validation with the ECMWF model and the radiosonde observations.

### 3.2.1 Representativeness error $\sigma_r$

To achieve an estimate of the representativeness error, COSMO-DE analyses of different seasons of the year 2016 are used. Although the Aeolus wind observations correspond to line measurements, the ECMWF and the DWD global model are treating the HLOS winds as point measurements. The radiosonde observations can also be regarded as single point measurements.

To get an estimate for the model representativeness error $\sigma_{r\_model}$, differences between a point and a line measurement (90 km for Rayleigh winds and Mie winds till 5 March 2019, 10 km for Mie winds after 5 March 2019) are determined, averaged over height levels according to the Aeolus range bin setting, and weighted by the mean number of Aeolus winds measurements for the Rayleigh and the Mie channel (Figure 2 right-hand side). Because the Aeolus HLOS winds mainly correspond to the zonal wind component, the calculations are only done for $u$. The resulting $\sigma_{r\_model}$ is 0.50 m s$^{-1}$ for the Rayleigh winds, 0.52 m

s$^{-1}$ for the Mie winds with 90 km horizontal resolution and 0.12 m s$^{-1}$ for the Mie winds with 10 km horizontal resolution.

For the estimation of the radiosonde representativeness error $\sigma_{r\_RS}$, it is necessary to make a distinction between the radiosondes for which the drift is assimilated (87 %), and those reports which only contain the launch position and time (13 %). For both cases, the temporal and the spatial part of the representativeness error, resulting from the collocation criteria, has to be considered. The error due to the spatial displacement is assessed by determining the differences between a point and a line

measurement as the weighted mean over distances up to 120 km in east-west and north-south direction, and calculating the weighted average over altitude. To consider the temporal displacement, a time offset value is estimated by assessing the representativeness error of the appropriate spatial displacement. The mean wind velocity over the validation period (15.27 m s$^{-1}$)





and the temporal collocation criteria of 90 min gives a spatial displacement of 82 km, which corresponds to a representativeness error of 1.26 m s$^{-1}$ for both channels with 90 km horizontal resolution and 1.40 m s$^{-1}$ for the Mie winds with 10 km

horizontal resolution. For the 13 % of the radiosonde data without the drift information, additionally an error component due to the spatial displacement up to 50 km and an error component due to the temporal displacement during the radiosonde ascents up to 90 min has to be considered. For the 87 % of the radiosondes with the drift information a temporal offset value for the 15 min time interval, into which the data are grouped has to be taken into account. Those parts of the representativeness error are calculated according to the parts resulting from the collocation criteria, using the COSMO-DE analysis. Compared to the high-

resolution ICON-LEM simulation, the COSMO-DE analysis underestimates the representativeness error. Figure 2 (left-hand side) shows the differences between a point and a line measurement averaged and weighted over distances up to 200 km as a function of altitude for the ICON-LEM and COSMO-DE data of the same date. On average, the offset value between the two models is 0.20 m s$^{-1}$. The offset value is added to the sum of the variances of the different error components, resulting in a representativeness error of 2.48 m s$^{-1}$ for the Rayleigh winds, 2. 49 m s$^{-1}$ for the Mie winds with 90 km horizontal resolution and 2.66 m s$^{-1}$ for the Mie winds with 10 km horizontal resolution.

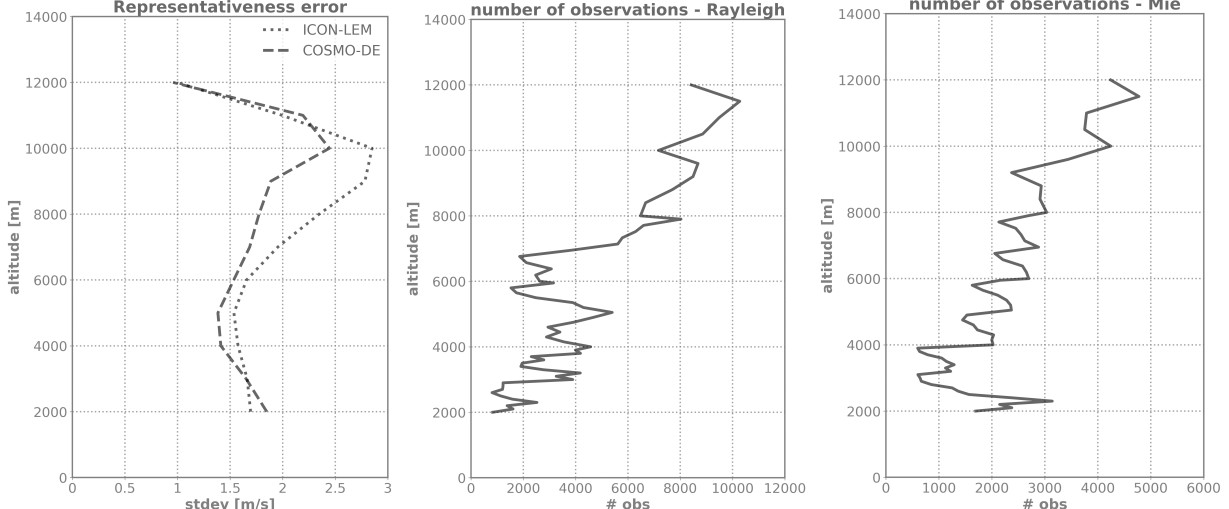

**Figure 2.** Left: Representativeness error estimated with differences between a point and a 90 km line measurement as a function of altitude for an ICON-LEM simulation (dotted line) and COSMO-DE analyses (dashed line). Right: the height profile of the mean number of measurements for the Rayleigh and the Mie channel.


### 3.2.2    Model error $\sigma_b$ and radiosonde wind observational error $\sigma_{RS}$

The ECMWF model error is derived from the ensemble data assimilation first guess error, stored in the ODB. It provides a good measure for spatial and temporal variation of the background error. Table 2 displays the values of $\sigma_b$ as mean over the validation period for the Rayleigh winds, and as mean over the time periods before and after the change of the horizontal





resolution for the Mie winds. They are determined for the latitudinal band between 23.5 and 65°N, and globally. The values taken for the model error are only valid at the start of the 4D-Var window. They are increasing during the 12-hr window. As NWP models in general show higher uncertainties in cloudy than in clear sky conditions, $\sigma_b$ is larger for the Mie winds.

The radiosonde observational error $\sigma_{RS}$ is assumed to be 0.7 m s$^{-1}$, according to the estimated GCOS (Global Climate Observing System) Reference Upper-Air Network (GRUAN) measurement uncertainty (Dirksen et al., 2014) .

### 285  3.2.3   Aeolus wind instrumental error $\sigma_{Aeolus}$

The Aeolus wind instrumental error is calculated using Equation 6a and Equation 6b. Table 2 shows the values of $\sigma_{Aeolus}$ for the validation with radiosonde observations and ECMWF model equivalents for the latitudinal band between 23.5 and 65°N for the Rayleigh and Mie winds, separated for the ascending and descending orbit phase. Additionally, $\sigma_{Aeolus}$ is derived for the global statistic using the ECMWF O-B values. The Rayleigh winds error estimate is 4.37 m s$^{-1}$ (4.23 m s$^{-1}$) for

the ascending (descending) orbit using radiosonde observations as reference data, and 4.07 m s$^{-1}$ for the ascending and the descending orbit for the comparison with model equivalents of the ECMWF model. The estimated error of the Mie winds with 90 km (10 km) horizontal resolution is around 2.68 m s$^{-1}$ (3.00 m s$^{-1}$) for the radiosonde validation, around 2.00 m s$^{-1}$ (2.75 m s$^{-1}$) for the model validation. For both channels $\sigma_{Aeolus}$ shows good agreement between the ascending and descending orbit phase. The differences between the model and radiosonde validation are at most 0.31 m s$^{-1}$, except for the

Mie winds with 90 km resolution. Since the representativeness error and the model error estimates have to be treated carefully, the discrepancies between the radiosonde and model validation are assumed to be in the range of the estimation uncertainties. Comparing the globally derived Aeolus winds instrumental error with the results of the validation statistics of the northern hemisphere, smaller values occur for the Mie winds, whereas the Rayleigh winds instrumental errors show good accordance. It has to be taken into account that the representativeness error, considered for the global statistics is based on a domain only

covering Central Europe. The results for the radiosonde and the model validation are found to correspond well to the results of Witschas et al. (2020) for comparisons with a 2-$\mu$m DWL during the validation campaigns WindVal III and AVATARE (Aeolus Validation Through Airborne Lidars in Europe) over Europe in late autumn 2018 and early summer 2019. By excluding the 2-$\mu$m DWL measurement error a Aeolus instrumental error of 3.9 - 4.3 m s$^{-1}$ (2.0 m s$^{-1}$) for the Rayleigh (Mie) winds is determined (Witschas et al., 2020). Rennie and Isaksen (2020) estimates the Aeolus instrumental error using the ECMWF

model on a global base by subtracting a background $u$ wind error of 1.6 m s$^{-1}$, resulting in a $\sigma_{Aeolus}$ of 4 - 5 m s$^{-1}$ (3 m s$^{-1}$) for the Rayleigh (Mie) winds. The slight discrepancies are probably related to the small selected regions around radiosonde collocation points, from which the validation results in Table 2 are derived. The global statistic in this study is based on a similar approach using restricted regions and short time periods. These limited areas are used in particular to avoid that the estimate of the random error is influenced by horizontal and temporal fluctuations of the bias.



**Table 2.** Overview of the estimated Aeolus winds instrumental error $\sigma_{Aeolus}$ and the single components of the calculation (Representativeness errors $\sigma_{r\_RS}$ and $\sigma_{r\_model}$, radiosonde observational error $\sigma_{RS}$, ECMWF model error $\sigma_b$ and random differences from the validation $\sigma_{val\_RS}$ and $\sigma_{val\_model}$) for the Rayleigh and Mie winds for the ascending and descending orbital pass for the northern hemisphere $(23.5 - 65°\text{N})$, restricted to the radiosonde collocations, and for a global statistic using the ECMWF model.

| | | | Rayleigh | | Mie (90 km) | | Mie (10 km) | |
| --- | --- | --- | --- | --- | --- | --- | --- | --- |
| | | | ascending | descending | ascending | descending | ascending | descending |
| Validation with radiosonde observations | | $\sigma_{r\_RS}$ | 2.48 m s$^{-1}$ | 2.48 m s$^{-1}$ | 2.49 m s$^{-1}$ | 2.49 m s$^{-1}$ | 2.66 m s$^{-1}$ | 2.66 m s$^{-1}$ |
| | | $\sigma_{RS}$ | 0.70 m s$^{-1}$ | 0.70 m s$^{-1}$ | 0.70 m s$^{-1}$ | 0.70 m s$^{-1}$ | 0.70 m s$^{-1}$ | 0.70 m s$^{-1}$ |
| | | $\sigma_{val\_RS}$ | 5.07 m s$^{-1}$ | 4.95 m s$^{-1}$ | 3.78 m s$^{-1}$ | 3.67 m s$^{-1}$ | 4.09 m s$^{-1}$ | 4.05 m s$^{-1}$ |
| | | $\boldsymbol{\sigma_{Aeolus}}$ | **4.37 m s$^{-1}$** | **4.23 m s$^{-1}$** | **2.76 m s$^{-1}$** | **2.60 m s$^{-1}$** | **3.03 m s$^{-1}$** | **2.97 m s$^{-1}$** |
| Validation with ECMWF model | | $\sigma_{r\_model}$ | 0.80 m s$^{-1}$ | 0.81 m s$^{-1}$ | 1.02 m s$^{-1}$ | 1.05 m s$^{-1}$ | 1.15 m s$^{-1}$ | 1.11 m s$^{-1}$ |
| | | $\sigma_b$ | 0.50 m s$^{-1}$ | 0.50 m s$^{-1}$ | 0.52 m s$^{-1}$ | 0.52 m s$^{-1}$ | 0.12 m s$^{-1}$ | 0.12 m s$^{-1}$ |
| | | $\sigma_{val\_model}$ | 4.18 m s$^{-1}$ | 4.18 m s$^{-1}$ | 2.19 m s$^{-1}$ | 2.43 m s$^{-1}$ | 2.96 m s$^{-1}$ | 2.99 m s$^{-1}$ |
| | | $\boldsymbol{\sigma_{Aeolus}}$ | **4.07 m s$^{-1}$** | **4.07 m s$^{-1}$** | **1.87 m s$^{-1}$** | **2.13 m s$^{-1}$** | **2.72 m s$^{-1}$** | **2.77 m s$^{-1}$** |
| Global statistic with ECMWF model | | $\sigma_{r\_model}$ | 0.83 m s$^{-1}$ | 0.84 m s$^{-1}$ | 1.23 m s$^{-1}$ | 1.21 m s$^{-1}$ | 1.32 m s$^{-1}$ | 1.30 m s$^{-1}$ |
| | | $\sigma_b$ | 0.50 m s$^{-1}$ | 0.50 m s$^{-1}$ | 0.52 m s$^{-1}$ | 0.52 m s$^{-1}$ | 0.12 m s$^{-1}$ | 0.12 m s$^{-1}$ |
| | | $\sigma_{val\_model}$ | 4.33 m s$^{-1}$ | 4.25 m s$^{-1}$ | 2.05 m s$^{-1}$ | 2.08 m s$^{-1}$ | 2.85 m s$^{-1}$ | 2.82 m s$^{-1}$ |
| | | $\boldsymbol{\sigma_{Aeolus}}$ | **4.22 m s$^{-1}$** | **4.14 m s$^{-1}$** | **1.56 m s$^{-1}$** | **1.62 m s$^{-1}$** | **2.52 m s$^{-1}$** | **2.50 m s$^{-1}$** |

## 4 Investigation of the Aeolus L2B HLOS Rayleigh wind bias

The following part concentrates on the Aeolus L2B HLOS Rayleigh winds bias. On a global scale, bias dependencies are investigated for different time periods, and accordingly, correction schemes are tested.

### 4.1 Rayleigh wind bias dependence on latitude and orbit phase

Figure 3 displays the Rayleigh winds bias as a function of latitude for the ascending and descending orbit phase. The values are binned into $10°$ latitude bins. Results are shown for March 2019 (Figure 3 **(a)**) and August 2019 (Figure 3 **(b)**). As in the validation statistics for the northern hemisphere, shown in Section 3, the two NWP models correspond very well along the climate zones. Largest differences appear in the tropics and subtropics. The comparison of Aeolus winds with inhomogeneously distributed radiosonde measurements overall shows good agreement as well. Outliers, as around $20°$ S or $80°$ N, are mainly related to small sample sizes.

Representative for winter and spring, Figure 3 (a) shows that the bias is quite constant with latitude in that period. Small differences between the orbital passes occur in the southern hemisphere and in the subtropical region of the northern hemisphere. From $40°$ N up to the north pole, almost no deviation between ascending and descending orbit is visible.

In August 2019 (Figure 3 **(b)**), the bias varies with latitude with an amplitude of 4 - 5 m s$^{-1}$. As seen in Section 3 for the





summer and autumn season, large differences between the orbit phases exist, in particular outside of the tropics. Around the
equator, the sign of the bias is positive for the ascending and descending orbit. Between the subtropical region and the poles,
the descending orbit bias is still positive, whereas the bias of the ascending orbit has a negative sign.

The results suggest that the satellites orbit phase and latitude position as well as the season seem to influence the Aeolus
Rayleigh winds bias. As the formulation of most data assimilation schemes assumes unbiased incoming observations, the cor-
rection of systematic differences is crucial. Thus, it is tested, if a bias correction approach as a function of latitude based on first
guess departures of the preceding week, separately for ascending and descending orbit, can remove the systematic differences
for the validation period (see Section 4.1.1).

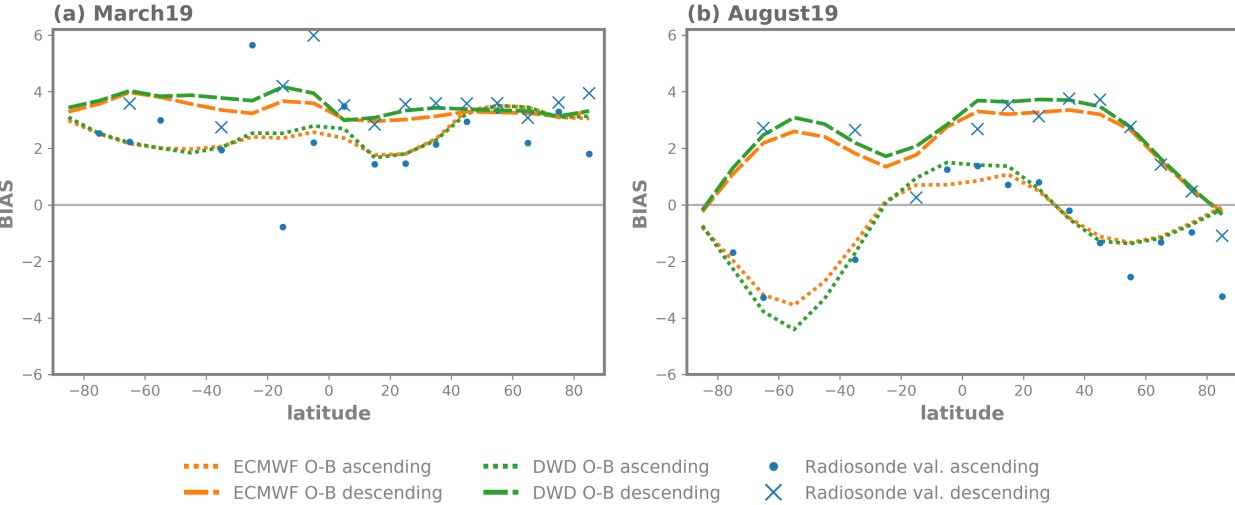

**Figure 3.** Aeolus HLOS winds bias as a function of latitude for ascending (dotted line) and descending (dashed line) orbit, calculated with
model equivalents of the ECMWF (orange) and the ICON model (green). In blue (point markers: ascending, cross markers: descending)
comparison results with collocated radiosonde observation are shown. Values are binned into latitude bins of $10°$. **(a)**: March 2019; **(b)**:
August 2019.

### 4.1.1   Rayleigh wind bias correction approach as function of latitude

Based on the previous results, a bias correction approach is evaluated and tested with the ECMWF IFS and the ICON model
monitoring data sets. For latitude bins of $10°$, the first guess departures from the previous seven days are averaged using the
following weights:

$$w_i = \frac{\frac{1}{1+i}}{\sum_{j=1}^{7}\left(\frac{1}{1+j}\right)}, i = 1, ..., 7 \tag{7}$$





with i=1 being the current day. The resulting correction values are subtracted from the first guess departure of the considered day and the residuals are averaged for each month of the validation period (Figure 4). Considering the effect of the orbit phase differences, this is done separately for the ascending and descending satellite pass. To estimate if the model bias matters three

different configurations are tested, which differ regarding the correction values: the bias correction values are based on the same model (dark filled markers); the bias correction value is calculated with the other NWP model (unfilled markers); the bias correction value is an average value of the two NWP models (light filled markers).

After applying the bias correction, a temporal variation as seen in Section 3 for the systematic differences is still apparent in the residuals. At the beginning of the Aeolus mission, the correction is quite efficient. In spring 2019, when the latitude dependence

is comparably weak and the bias comparably high, a residual up to over $1 \text{ m s}^{-1}$ remains. After the processor update in May 2019, when the Rayleigh ascending wind bias tends to be negative, also the residual bias exhibits a negative sign. Differences between the two models regarding the sign of the remaining bias are visible in September 2018 for the ascending orbit and in December 2019. In total, the correction is able to clearly decrease the systematic differences, but there is a remaining bias, in particular in phases with large temporal changes of the bias. The seasonal variation of the bias and the influence of the

latitudinal position of the satellite suggest a link to temporal and spatial variations in long wave and solar radiation. Including the longitudinal component, the spatial bias dependence for different time periods is examined in more detail in the following Section 4.2.

Table 3 presents the mean absolute residual bias averaged over the validated time period for the three applied latitude dependent correction values. In total, the bias is reduced by almost $1 \text{ m s}^{-1}$ for the DWD global model and even more than $1 \text{ m s}^{-1}$ for the

ECMWF model. A correction based on the previous seven days of the own model yields a comparable mean absolute residual bias for the ECMWF IFS and the ICON model. Correcting the ECMWF IFS model with the correction values calculated with the ICON model gives overall the smallest remaining bias and largest reduction. The ICON model O-B statistic in contrast shows worse results when applying information of the ECMWF IFS model to correct for the latitude dependent error. However, the differences between the three configurations are small, which again indicates that model biases don't have a dominant effect

on the bias assessment.

Altogether, these results show that a temporally varying latitude dependent bias is present for the L2B Rayleigh wind product. Results from the evaluation with the two independent NWP models and in situ observations are overall in good agreement. A latitude dependent bias correction successfully reduces the bias, but on average, a bias of 0.37 - 0.59 $\text{m s}^{-1}$ remains. The remaining bias is related on one hand to phases with temporal changes of the bias and on the other hand on longitudinal

differences that are investigated further in the subsequent section.

## 4.2 Rayleigh wind bias dependence on longitude, latitude and orbit phase

Figure 5 shows two-dimensional plots of the Aeolus Rayleigh HLOS winds bias for January, May, and September 2019 for the ascending and descending satellite orbit. In January, when the orbit phase dependence is less pronounced, small fluctuations with longitude and latitude are visible in the tropical and subtropical regions. Large positive bias values occur between 30 and

90°N, mainly for the ascending orbital pass, and in the tropics, more present for the descending orbital pass. The band of larger





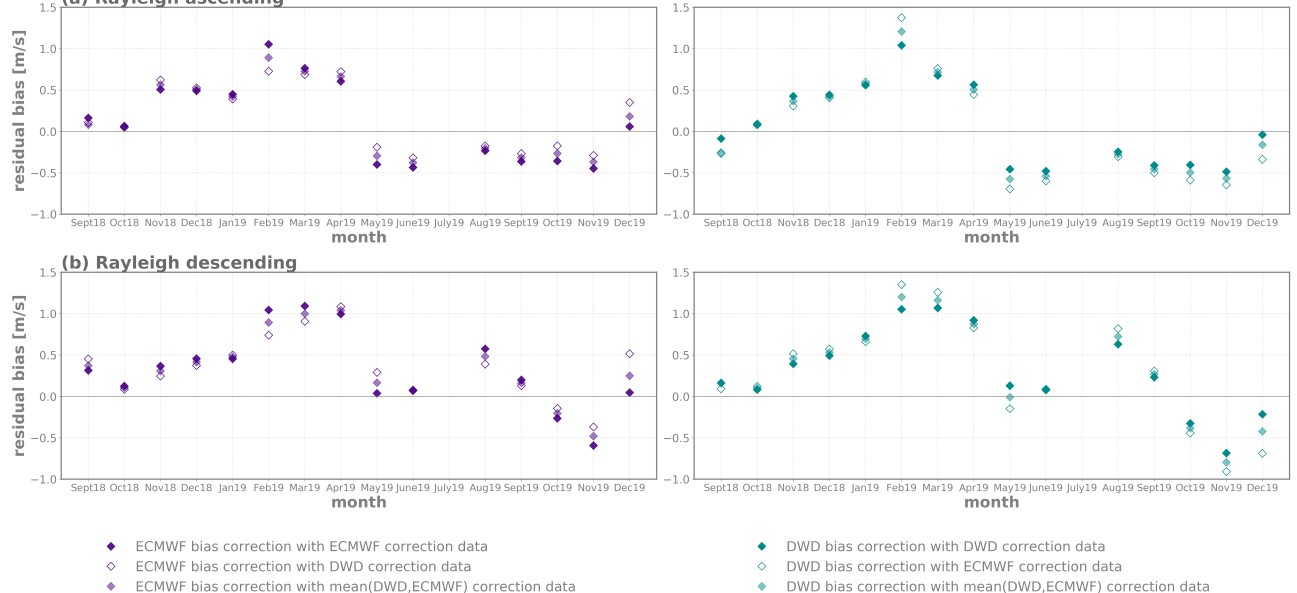

**Figure 4.** Residual after a latitude dependent bias correction, separately for Rayleigh ascending **(a)** and descending **(b)** orbit averaged over one month. On the left (violet) the ECMWF model residuals, on the right side the ICON model residuals (cyan) are displayed. The correction values are either based on the previous week of the model equivalents of the own model (dark filled markers) or the other NWP model (unfilled marker) or on an average value of both models (light filled markers).

**Table 3.** Mean absolute residual bias of the ECMWF and the ICON model after a latitude dependent bias correction for three different configurations for the time period from September 2018 to end of December 2019.

|  | ascending | | descending | |
|---|---|---|---|---|
|  | ECMWF | DWD (ICON) | ECMWF | DWD (ICON) |
| without bias correction | 1.41 m s$^{-1}$ | 1.28 m s$^{-1}$ | 1.64 m s$^{-1}$ | 1.54 m s$^{-1}$ |
| correction value based on ECMWF model | 0.43 m s$^{-1}$ | 0.53 m s$^{-1}$ | 0.44 m s$^{-1}$ | 0.59 m s$^{-1}$ |
| correction value based on DWD (ICON) model | 0.37 m s$^{-1}$ | 0.43 m s$^{-1}$ | 0.42 m s$^{-1}$ | 0.48 m s$^{-1}$ |
| correction value based on $\overline{\text{(ECMWF, DWD )}}$ | 0.39 m s$^{-1}$ | 0.48 m s$^{-1}$ | 0.43 m s$^{-1}$ | 0.52 m s$^{-1}$ |





systematic differences found in the tropics seems to match with the Intertropical Convergence Zone (ITCZ), which moves further south from the equator during the southern summer. In May, the orbit phase dependence of the systematic differences is more distinct. For the ascending orbit, longitude fluctuations of large negative bias values over land appear in the temperate and polar areas of the northern hemisphere. Variability is also still present in the equatorial region. When the satellite moves
from north to south these tropical fluctuations are less conspicuous. Except for the polar region of the northern hemisphere, the bias is mostly positive with highest values between 30 and $90°$S. The three gaps on the southern hemisphere around $60°$S are due to a technical issue at ECMWF. In autumn, when latitude and the satellite's orbit phase influences the systematic error most, also a significant longitude dependence is apparent. The land sea fluctuations for the ascending orbital pass on the northern hemisphere and in the tropical region are more pronounced. For the descending orbit, variability is mainly present
in the southern hemisphere and it is not clear whether this is linked to the land sea distribution. The positive bias band in the ITCZ region is still present for both orbits.

Furthermore, the results of the ECMWF IFS model are again compared to the ICON model O-B statistics (Figure 6), showing overall good agreement. Larger differences only emerge in the tropics, the area where NWP models in general differ the most, and in the midlatitudes of the summer hemisphere.

Figure 5 highlights that in addition to the satellites flight direction, the latitude, and seasonal variations also longitudinal fluctuations affect the Aeolus measurements systematically, supporting the assumption that radiative effects play an important role. To examine the extent of the influence of the longitude component, the bias correction approach outlined in Section 4.1.1 is repeated taking both geographical dimensions into account (see Section 4.2.1).

### 4.2.1   Rayleigh wind bias correction approach as function of latitude and longitude

For the ECMWF model, a two-dimensional bias correction approach is tested using the previous seven days of Aeolus HLOS O-B statistics as a function of latitude and longitude averaged and weighted (Equation 7). Bin sizes are chosen to be $10°$ for both, latitude and longitude. To also consider the seasonal variation, Figure 7 displays the residuals (rose cross markers) averaged for each month for the whole validation period for the ascending and descending orbit. To get an impression of how strong the longitudinal bias variation is, the results are compared to the one-dimensional latitude dependent correction approach
from Section 4.1.1. The mean absolute remaining bias for both correction formulations is provided in Table 4. Altogether, the residual has been decreased by almost 50 % when considering the longitude dependence for both satellite orbit passes. Main improvements occur for the bias correction in late winter and early spring 2019, where a one-dimensional correction approach isn't that effective. Right after the mission's start, in May 2019 and at the end of the year the remaining bias is increased when taking the longitudinal dimension into account. In these months, the one dimensional latitude dependent correction approach
almost removes the systematic differences already.

A discussion on possible reasons for the systematic bias variations and a summary of the findings in this study are presented in the final Section 5.



**Figure 5.** Aeolus Rayleigh HLOS winds bias determined with O-B statistics of the ECMWF model as a function of latitude and longitude for January 2019 (top), May 2019 (center) and September 2019 (bottom) - please note that a different wind speed range is used for the color scales. On the left for the ascending orbit, on the right for the descending orbit.





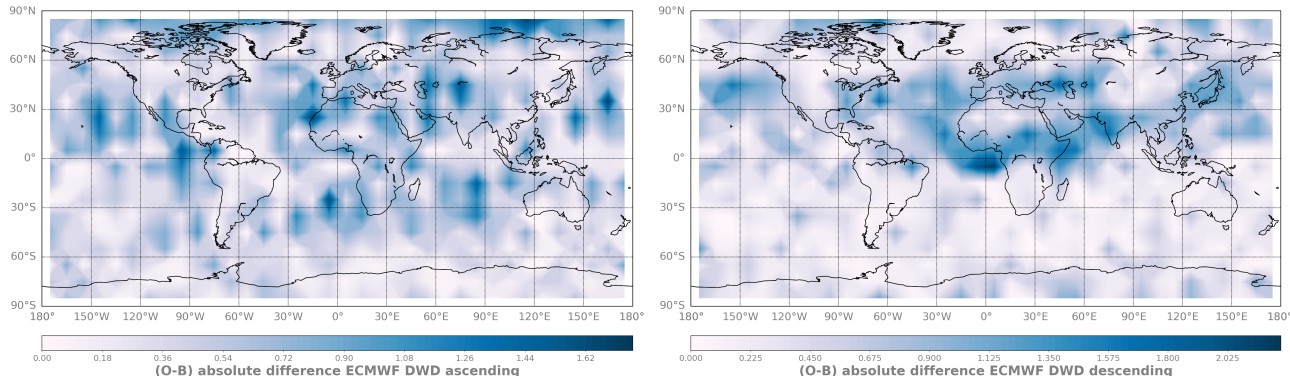

**Figure 6.** Absolute differences between the ECMWF IFS and the ICON model O-B for May 2019. On the left for the ascending orbit, on the right for the descending orbit.

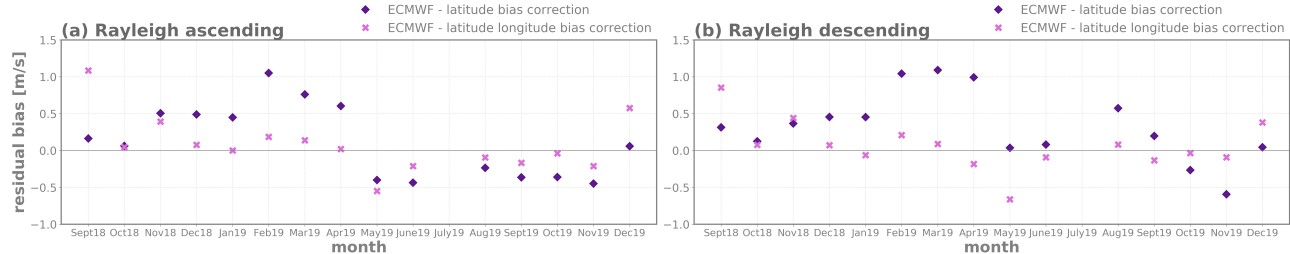

**Figure 7.** Residual after a latitude dependent bias correction (magenta diamond marker) and a two dimensional latitude-longitude dependent bias correction (rose cross marker) averaged over one month, using the ECMWF model equivalents. On the left for Rayleigh ascending **(a)** and on the right for descending **(b)** orbit phase.

**Table 4.** Mean absolute residual bias after a latitude and a latitude-longitude bias correction approach using the ECMWF model for the time period from September 2018 to end of December 2019.

| type of bias correction | ascending | descending |
|---|---|---|
| latitude | 0.43 m s$^{-1}$ | 0.44 m s$^{-1}$ |
| latitude - longitude | 0.25 m s$^{-1}$ | 0.23 m s$^{-1}$ |



## 5 Summary and discussion

This study provides an overview of validation activities to determine the Aeolus HLOS wind errors and to understand the biases
by investigating possible dependencies. To ensure meaningful validation statistics, collocated radiosondes and two different
global NWP models, the ECMWF IFS and the ICON model of DWD, are used as reference data.

Overall, the determined mean wind differences of the comparisons with all three reference data sets show good concordance.
This confirms that the detected bias is due to Aeolus L2B systematic wind errors and not the reference data set. A time series
demonstrates that the Aeolus winds systematic differences vary considerably during the time period from the satellite's launch
until the end of December 2019 (Section 3.1). Further, there are differences in bias between the ascending and descending
orbit phase, which mainly occur for the Rayleigh channel in late summer and autumn. Whereas the Rayleigh descending phase
winds are positively biased in these months, the ascending phase shows negative bias values. The Mie winds are less biased in
total, but more fluctuating. The mean absolute bias is found to be approximately 1.8 - 2.3 m s$^{-1}$ for the Rayleigh winds and
1.3 - 1.9 m s$^{-1}$ for the Mie winds. These values are beyond the mission requirements of Aeolus, which states that the bias
should be smaller than 0.7 m s$^{-1}$ (ESA, 2016). However, it is demonstrated that the bias can be reduced to values lower than
the mission requirement through calibration with observations and model fields of the preceding week.

The random differences of the Rayleigh winds show temporal changes, that are mainly related to changes in the laser output
energy. The Mie winds random differences are less influenced by the laser energy and quite constant with time. The mean
scaled MAD of the comparisons shows the highest values when using the radiosonde observations as reference, which is
caused by representativeness errors. The NWP model scaled MAD is larger for the ICON model O-B statistics than for the
ECMWF first guess departures, likely due to the neglection of temporal changes within the assimilation window in the DWD
assimilation system. The Aeolus instrumental wind error $\sigma_{Aeolus}$ is estimated by determining the representativeness error for
the ECMWF model validation and the radiosonde comparison, and by taking the ECMWF model error and the radiosondes
measurement error into account. For the Rayleigh winds $\sigma_{Aeolus}$ is in the range of 4.1 - 4.4 m s$^{-1}$, for the Mie winds with 90
km horizontal resolution in the range of 1.9 - 2.8 m s$^{-1}$, and for the Mie winds with 10 km horizontal resolution in the range of
2.7 - 3.0 m s$^{-1}$. Given, that the representativeness and the model error estimate exhibit large uncertainties and the subtracted
bias varies a lot with latitude and longitude, these differences are probably within the range of the uncertainty of the estimate.
A global statistic using the ECMWF O-B values of limited areas (10° latitude x 10° longitude) shows only slightly smaller
values for the Mie winds instrumental error, whereas the global Rayleigh winds instrumental error is in good agreement with
the validation results based on the northern hemisphere.

The second part (Section 4) of the results of this study further investigates the Rayleigh wind bias and its dependencies. Besides
the satellite's flight direction and seasonal differences, also latitude and longitude influence the systematic differences. Again,
the good agreement between the different validation data sets raises confidence that the results are not influenced by issues of
the reference data sets. The latitude bias dependence and differences between the orbit phases mainly occur in late summer
and autumn in the subtropics and temperate climate zone. Since a one-dimensional latitude dependent correction approach is
reducing the bias, but still, a temporal trend of remaining bias values of 0.37 - 0.59 m s$^{-1}$ occur, it turned out that also the



longitude component should be taken into account. When the satellite moves north, longitudinal variations are especially found in the tropics and between 20 and 60°N, while for the descending orbit phase systematic differences mainly occur between 20 and 60°S. These variations suggest correlations with land-sea distribution and tropical convection. A latitude-longitude correction approach using the ECMWF model equivalents is able to reduce the systematic error to 0.23 - 0.25 m s$^{-1}$.


At ECMWF, as part of the Aeolus Data Innovation and Science Cluster (DISC), the dominant source of the Rayleigh wind bias issues have been explained. It was found that the bias is correlated with the temperature gradients across the ALADIN primary mirror M1 of the telescope (Rennie and Isaksen, 2020). The M1 mirror temperature variation in turn is related to varying short and long wave radiation of the top of the atmosphere and the mirror's on board thermal control in response to this, which explains the seasonal differences and the connection to features like convection and variations between land and sea. Since 20 April 2020 a M1 bias correction scheme has been applied operationally in the L2B processor, using a multiple linear regression method of all M1 telescope thermistors developed by the Aeolus DISC (Rennie and Isaksen, 2020). A re-processed data set including a M1 bias correction will be available in near future. This data set should decrease the Aeolus instrumental error estimate and differences between the model and radiosonde comparisons.


*Data availability.* Since May 2020, Aeolus data is publicly available at the ESA Aeolus Online Dissemination System.

*Author contributions.* AM performed the data analysis and prepared the main part of the publication. MW supervised the work. MW, OR, MR and AG contributed to the development of methods and analysis of the data. OR and AG communicated important information on the Aeolus data quality and processing. MR provided knowledge about the ECMWF feedback files and the Aeolus wind bias. AC provided ideas for the bias correction approach and information about the DWD monitoring data. All co-authors engaged in discussions and contributed to the interpretation of the results.


*Competing interests.* The authors declare that they have no conflict of interest.

*Disclaimer.* The presented work includes preliminary data (not fully calibrated/validated and not yet publicly released) of the Aeolus mission that is part of the European Space Agency (ESA) Earth Explorer Programme. This includes wind products from before the public data release in May 2020 and/or aerosol and cloud products, which have not yet been publicly released. The preliminary Aeolus wind products will be reprocessed during 2020 and 2021, which will include in particular a significant L2B product wind bias reduction and improved L2A radiometric calibration. Aerosol and cloud products will become publicly available by spring 2021. The processor development, improvement and product reprocessing preparation are performed by the Aeolus DISC (Data, Innovation and Science Cluster), which involves DLR, DoRIT, ECMWF, KNMI, CNRS, ST, ABB and Serco, in close cooperation with the Aeolus PDGS (Payload Data Ground Segment). The analysis has been performed in the frame of the Aeolus Scientific Calibration and Validation Team (ACVT).




*Acknowledgements.* This work was funded by the German Federal Ministry for Economic Affairs and Energy (BMWi) under grant no.(FKZ) 50EE1721A. We want to thank ESA for the provision of the preliminary Aeolus data products. We further thank the ECMWF and the DWD for providing monitoring data. The valuable discussions within the EVAA consortium (LMU, DWD, DLR) is greatly acknowledged as well.



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
