# Peer review of "Validation of Aeolus winds using radiosonde observations and NWP model equivalents"

_Atmospheric Measurement Techniques, 2020_

## Referee Comment (RC1) · Anonymous Referee #1 · 8 Nov 2020

Review

The manuscript "Validation of Aeolus winds using radiosonde observations and NWP model equivalents" by Martin et al. provides a good examination of the Aeolus data, mainly for northern midlatitudes but also globally. The validation results cover Aeolus data from its launch to the end of December 2019. Thus, it provides a useful analysis for users of the data. Additionally, it is demonstrated how the bias in Aeolus data can be reduced bringing the data quality towards the mission requirements. The manuscript is well written, concise and has appropriate figures. Only one minor comment is listed below. The reviewer recommends publication after a minor revision.

Minor comment: - paragraph 4.1.1: In Figure 4, the global average is shown. It would be of interest to see the latitudinal behavior. This could be done as a shaded plot

with time and latitude on the axis for only one correction per model.The three different choices shown only as average numbers over the time series in table 3 is sufficient, but needs a significance test for the diffference between methods.

Technical stuff> - Check spelling for the references: Källén, Zagar (with the accent over the Z), and Savli (with accent over S, first name is Matic) - line 118: Consider to rephrase "during the ascent and the ascent time" - line 142: typo "operationally" - generally: Terms like "the Rayleigh winds mean absolute bias" seem unusually to me. Consider just "the Rayleigh wind mean absolute bias". And equivalently for Mie and random errors. - line 185: typo "differs" - line 188: Does the lower representativeness error of the radiosondes mean better agreement with Aeolus? Please clarify. - line 189: Higher resolution of radiosondes? Consider. - line 197: Please rephrase the sentence: "For the Mie winds, the global...". - line 201: Incompatible to what? Consider to remove term "incompatible". - line 202: typo "positively" - line 206: "mean differences" = bias? Please keep consistent names. Also in in the rest of the paragraph. - line 225: "random difference" = random error, standard deviation, sigma? Same as previous comment. - line 227: typo "shows" and "but" - line 230: consider "signal does not only depend on" - line 232: For the comparison of the standard deviation and MAD, you could show a histogram. This could also be used to justify the thresholds for the gross errors in line 106 and 107. - line 304: typo "estimate" - line 359: typo "do not" - line 398: typo "is not" - Figure 5: Please add the month name into the panels of the figure. - line 414: typo "state" - line 435f: Consider to rephrase the sentence "Since a one-dimensional...". It is difficult to understand. - line 444: typo "onboard" - references: Correct also names from first point.
* * *

---

## Referee Comment (RC2) · Anonymous Referee #2 · 9 Nov 2020

General comments:

The manuscript is well structured and addresses an important topic of the new observation system which is of significant importance for the NWP. It is found as an important contribution to the Aeolus special edition and is well in the scope of the scientific journal. The methods applied are well designed. It is appreciated that a reference data from two independent models are provided, which allowes to address the issue of the model related bias as current bias correction methods are still dependent on the model itself. The estimate of the various error sources is explained, however, the section on the representativeness error is not very clear (please see next paragraph for details). Below I list several minor issues that should be resolved. Some suggestions are as well provided.

[Figure]

Specific comments:

Abstract is overall well understood. However, I have some questions: The bias correction applied in April 2020 is the M1 temperature correction, however, in the abstract the longitute-latitude related bias correction is described. I found this confusing when reading the abstract and I suggest modifying sentence on line 20 such that it is clear that the operational bias correction at ECMWF is not the one studied in the manuscript.

What does expression "analytical" (Line 49) represent? I don't see it necessary and I suggest removing it.

I believe in methodology section it should be mentioned what is the time span of the validation (i.e. validation period). It is mentioned in terms of baseline but not in terms of date-time, which is important for readers not directly related with the Aeolus mission.

A word on quality control. This is first mentioned in (Line 106). But it is confusing because it is not clear what all is part of quality control? Is it defined just by L2B HLOS error estimate or anything else? There were problems with hot pixels in FM-A period, which has been handled with specific QC (for example). I suggest adding few sentences about this in case more complex QC has been used (in methodology).

The sentence given in lines (145-147) is not clear. What is the meaning of ". . . 10 deg. Latitude x 10 deg. Longitude and limited periods of 7 day"? I am not sure how to interpret this information. Could this be better explained?

In the collocation methodology of radiosondes. What is the reason for choosing the 500 m in vertical (Line 123)? Is this some sort of trade-off to have enough large sample? The HLOS range-bins are thinner near the ground especially for Mie.

For the model description given in Section 2.3 I suggest adding the information on horizontal resolution of both models (like it is provided for COSMO and ICON LES in the text below). I think this is important for the understanding the representativeness error of HLOS winds when compared to various models.

The methodology on the estimation of representativeness error is not very clear (section 2.4). The same for results give in Section 3.2.1. It is not clear how the estimation of the various representativeness errors is actually performed (for the purpose of reproducibility of data presented in manuscript). How exactly are two high resolution models used to estimate the representative error of HLOS if first L2B HLOS is compared to global model (Eq. 6b) and second to radiosondes (Eq. 6a)? What is a definition of point and line measurement in the scope of estimation of representativeness error (Line 250)? I don't understand how values are computed (Line 254, 255, 264). Values reported in Lines 254,255 are found as sigma_b in Table 2. Please clarify. I suggest providing more detailed explanation of the methodology used (in section 3.2.1) in section 2.4?

In line 183 (section 3.1 in general) and Table 1 I found some confusing information about the term "mean absolute difference" (and mean absolute bias). What exactly is computed here? Is this an average of absolute values of BIAS shown in Figure 1. If so, in Table 1 |BIAS| is confusing?

At the end of Line 195: " ... of limited areas (10 deg. Latitude ...". This is related to the comment given before. I don't understand what is meant by this. How are these limited areas used for computation of global statistics? (See please the same comment above).

Line 197. I would not agree for Mie descending. But I would say that the values for global statistics are in the range of values of of all three (radiosonds, ECMWF and DWD) local statistics . Which, on the other hand, does not hold for Rayleigh-clear statistics.

Line 218. Which bias correction update ? I suggest being specific and to mention that this is the one that is validated in the following sections (the lon-lat dependent one).

In Table 2 the sigma_b for Mie on 90 km is about 0.5 m/s but only 0.12 m/s when on 10 km. What is a scientific reason for this? Because no matter the Mie accumulation

the IFS HLOS is always an interpolation to the L2B HLOS center of gravity location. So is this the result of QC used or maybe better L2B classification on clear cloudy for smaller accumulations?

Line 295: Sentence started with "Since the representativeness ..." is not clear. What does it mean? What is the range of estimated uncertainties?

Line 337: It is confusing how the bias correction is applied (Eq. 7). "i=1 being the current day" suggests that at the day of applying the bias correction the O-B for that same day are as well included. Isn't that the bias correction factor for a particular day is provided from O-B of previous days?

Figure 6: the pattern looks very random. Are these differences statistically significant (especially for ascending orbit)?

How does the computed estimate of the Aeolus instrument error compare with the L2B estimated HLOS instrument error? By the values reported in the manuscript this seems quite well in range, but I miss this comparison. It would be of interest to see how good is the L2B error estimation algorithm. Has this been studied?

In addition, for Conclusions. The bias correction method used is essentially a temporal and spatial smoothing. This is as well a source of bias residuals shown in Figure 4 and 6, as bias correction method is not able to react on fast changes in systematic errors that happens along the Aeolus orbit.

Technical corrections:

A general note on Figures. I suggest to use the same framework on all images. This is by using labels (a,b ...). At the moment this is not true everywhere and on some images (Right, Left, ...) is used instead. In addition on some Figure there is no units, this should be corrected.

Line 8: instead of "comparisons" it would be maybe better to use "the validation".

Line 11: If the "independent reference data sets" are the two models, I suggest here to be specific and change this with ". . . between the two model data sets" or similar.

Line 11: I suggest to switch "representativeness" into "representative of Aeolus winds" .

Line 13: Sentence (To achieve . . .) is not well understood. I suggest turning it around "Besides the . . . the Aeolus instrument error is estimated . . ." or similar.

Line 16: The expression "depend on" should probably be changed to "vary with".

Line 25: Better be specific. I suggest to switch "Earth's wind" to "atmospheric wind".

Line 26: The sentence "Within seven days, . . ." is not completely correct. The orbit does not cover the globe as the footprint is in scale of meters and orbits are separated by 1000 km or more. But the repeat cycle of the orbit is about 7 days. I suggest to rewrite this.

Line 27: Could remove the "only one large instrument", it is not necessary.

Line 31: First two sentences could be switched. It reads better.

Line 36: Could remove "on" and "parts"

Line 42: At the end of the sentence ". . . geostrophic balance" I suggest to put some reference. For example https://doi.org/10.1175/BAMS-86-1-73

Line 45,46: I suggest to add few additional valuable references (https://doi.org/10.1256/qj.05.83, https://doi.org/10.1002/qj.43, https://doi.org/10.1002/qj.2430).

Line 67: I suggest to put a new paragraph just before "The text is struct. . ."

Line 71: "representativeness error of comparisons", this is very confusing. Could the expression "comparisons" be removed here?

Line 80: In reference "ECMWF: . . ." the "ECMWF:" should be removed.

Line 86: "The resulting HLOS wind" should be modified by "The resulting HLOS wind observation therefor represents a horizontal average over ..."

Line 87: The reference "Matic et al" should be changed to "Šavli et al"

Line 87 (end): "the" should be removed.

Line 89: "..., to avoid systematic errors..." should be changed to " ... to reduce the HLOS systematic error ..." (for example). This is because there are other sources of systematic error that exist.

Line 93: I suggest to change the "... types" to "... resulting in four wind products (i.e. Rayleigh-clear, Rayleigh-cloudy, Mie-clear and Mie-cloudy)." to be more specific.

Line 95: Again the reference "EMCWF: ..." should be corrected by removing "ECMWF:"

Line 115 (and 116): I suggest to rewrite a bit into "...is taken into account by splitting data into groups of 15 min."

Line 116 (by the end): Please correct "contains" into "contain".

Line 118: Please correct "the ascent and the ascent" into " the ascent and the descent".

Line 156: I suggest to change the heading title into: "statistical measures", "statistical metrics", "validation metrics", or something similar. "Statistical characteristics" is not valid for description of statistical metrics that have been used.

For Eq. 2 I suggest to add that index "i" represents time.

Eq 3-4 seem incomplete. In results bias is provided for radiosonde, and 2 models. I suggest to explain here that different biases and STD are provided for different reference data. This should be seen by equations defined here. At the moment it seems that Eq 6 for STD is always computed taking BIAS from model (even if STD is computed for radiosondes). I believe this is not what has been used. Correct me please if

I am wrong.

Eq 5 is as well confusing. I suggest to replace this with the definition for MAD (so removing the "scaled"), which is a very basic measure. And in the text it should be mentioned that scaled MAD is used instead which is defined as 1.4826*MAD because it provides a measure similar to STD for normal samples.

Title of heading in section 3 is confusing. It suggest shorten it. Maybe something like "Aeolus HLOS error time series characteristics".

Line 199: Please modify ". . . wind bias, . . ." to "...wind bias and random error, . . ."

Line 204: "modes" is probably "mode"?

Line 227: Please correct "bu" to "but"

Line 237: Please correct "differences" to "difference"

Line 253: Please use the uniform Figure labels (i.e. a,b,. . .)

Figure 2: Please use labels a,b,. . .

Table 2: In caption "...(Representativeness errors . . ." should be corrected to " ...(representativeness errors ...)"

Line 329: "Thus, it is tested . . ." should be correct to "Thus, it is first tested . . .."

Figure 3: missing units.

Line 401: I think this is not needed as this is already given in the last paragraph of Introduction.

Figure 5: missing labels

Figure 6: missing labels

Line 406 (at the end): Please correct "as" to "as a"

Line 480, 483 and 485: Please check if this kind of referencing is correct.

Line 502: Please correct "Matic. S" to "Šavli. M" and "Zagar. N" to "Žagar. N"

Line 536: Please correct "Zagar. N" to "Žagar. N"

———————————————————

---

## Referee Comment (RC3) · Anonymous Referee #2 · 9 Nov 2020

I am sorry for a mistake in my initial review. I forgot to add that this is a "minor revision".
* * *

---

## Author Comment (AC1) · 17 Jan 2021

**Response to Referee Comment (RC1) on**

**Validation of Aeolus winds using radiosonde observations and NWP model equivalents**

*https://doi.org/10.5194/amt-2020-404*

We appreciate the referee's thoughtful and valuable review. The responses to the individual comments and the corresponding changes in the manuscript are presented in the following.

**General Comment:**

*The manuscript "Validation of Aeolus winds using radiosonde observations and NWP model equivalents" by Martin et al. provides a good examination of the Aeolus data, mainly for northern midlatitudes but also globally. The validation results cover Aeolus data from its launch to the end of December 2019. Thus, it provides a useful analysis for users of the data. Additionally, it is demonstrated how the bias in Aeolus data can be reduced bringing the data quality towards the mission requirements. The manuscript is well written, concise and has appropriate figures. Only one minor comment is listed below. The reviewer recommends publication after a minor revision.*

**Minor Comment:**

*paragraph 4.1.1: In Figure 4, the global average is shown. It would be of interest to see the latitudinal behavior. This could be done as a shaded plot with time and latitude on the axis for only one correction per model. The three different choices shown only as average numbers over the time series in table 3 is sufficient, but needs a significance test for the difference between methods.*

Response to the Minor Comment:

As requested, we performed a significance test and added, in that respect, the following sentence:

**Overall, the bias correction approaches show a statistically significant reduction in bias. However, no significant differences between the individual methods were found (following a Student's t-distribution), which again indicates that model biases do not have a dominant effect on the bias assessment.**

The plots below show the latitude dependence of the residual bias after the correction with preceding departure information, as suggested by the reviewer. However, we are not convinced that these plots really add significant information for the reader and we think that the interpretation of features on this plot would be largely speculative given a range of potential causes for a residual bias (various known and unknown fluctuations in the instrument performance; variations of model biases both in the preceding training data as well as in the verification data set; fluctuations of thermal emission from the earth based on the synoptic situation that may affect the instrument). We therefore prefer not to show this additional figure. The main intention was to show that the correction successfully reduces the bias, but does not eliminate it. This is already shown clearly by the average values.

[Figure]

[Figure]

***Further modification:***

The following reference has been added to the introduction part as example for already published Aeolus wind validation studies:

**Baars, H., Herzog, A., Heese, B., Ohneiser, K., Hanbuch, K., Hofer, J., Yin, Z., Engelmann, R., and Wandinger, U.: Validation of Aeolus wind products above the Atlantic Ocean, Atmospheric Measurement Techniques Discussions, 2020, 1–27, https://doi.org/10.5194/amt-2020-198, 2020.**

***Technical stuff***

*line 118: Consider to rephrase "during the ascent and the ascent time"*

> changed accordingly: "Due to the radiosonde drift during the sounding and the ascent time, additional errors arise. "

*line 185: typo "differs"*

> changed accordingly: differ --> "differs"

*line 188: Does the lower representativeness error of the radiosondes mean better agreement with Aeolus? Please clarify - line 189: Higher resolution of radiosondes? Consider*

> changed accordingly: "This can be explained by the larger representativeness errors associated with radiosondes, which can be regarded as in situ point measurements. Besides the higher spatial resolution of a radiosonde observation compared to the resolution of a global NWP model, ..."

*line 197: Please rephrase the sentence: "For the Mie winds, the global..."*

> changed accordingly: "For the Mie winds, the global statistic shows values in the range of the results of the three local validation statistics around the radiosonde collocations."

*line 201: Incompatible to what? Consider to remove term "incompatible"*

> no changes, as we think the current wording is appropriate. The Aeolus L2B data of different time periods are incompatible, due to different processor baselines (different physical principles) and calibration file updates.

*line 206: "mean differences" = bias? Please keep consistent names. Also, in in the rest of the paragraph.*

> See explanation of notions in Section 2.5.: Given that the model bias for long validation periods and large scales is usually small in comparison to that of Aeolus observations, the mean difference between the Aeolus observations and the reference data can be referred to as bias. --> no changes

*line 225: "random difference" = random error, standard deviation, sigma? Same as previous comment.*

> For the comparisons only the expression 'random differences' is used. To quantify the random differences, the standard deviation as well as the scaled MAD is used (explanation in Section 2.5.). For the estimation of the Aeolus instrumental error, the total variance of the differences between radiosonde observations and Aeolus HLOS winds $\sigma^2_{val}$ is described by the scaled MAD (see first part of Section 3.2), because it is more resilient to single outliers (see Section 2.5.). These terms are consistent in the paper. --> no changes

*line 230: consider "signal does not only depend on"*

> changed accordingly: "... signal does not only depend on..."

*line 232: For the comparison of the standard deviation and MAD, you could show a histogram. This could also be used to justify the thresholds for the gross errors in line 106 and 107.*

> changed accordingly: we interpret this comment to be basically related to the estimated error thresholds. Ongoing investigations show that the L2Bp estimated error has to be treated carefully due to a bug until the baseline change in April 2020. This issue is not fully understood yet and a specific justification of the used threshold would deserve a separate study. However, to point out how many data pass the quality monitoring used in this study, the following information is added to Section 2.1.:

**Based on a compromise between the quality of the data set and the number of observations that pass the quality control, Rayleigh winds with an estimated error greater than 6 m s⁻¹ and Mie winds with an estimated error greater than 4 m s⁻¹ are excluded. Thus, on average over the validation period about 70 % of the Rayleigh and 76 % of the Mie winds are available for the analysis.**

*line 435f: Consider to rephrase the sentence "Since a one-dimensional...". It is difficult to understand*

> changed accordingly: "A one-dimensional latitude dependent correction approach, based on the previous seven days, is able to reduce the bias, but still, a temporal trend of remaining bias values of 0.37 - 0.59 m s$^{-1}$ occur.

*line 444: typo "onboard" - references: Correct also names from first point*

> changed accordingly:  on board --> "on-board"

*Check spelling for the references: Källén, Zagar (with the accent over the Z), and Savli (with accent over S, first name is Matic)*

> changed accordingly: "Källén", "Šavli", "Žagar"

*line 142: typo "operationally" - generally: Terms like "the Rayleigh winds mean absolute bias" seem unusually to me. Consider just "the Rayleigh wind mean absolute bias". And equivalently for Mie and random errors*

> changed accordingly:  operational --> "operationally"
> generally, terms like 'Rayleigh/Mie winds mean absolute bias' are changed accordingly: "the Rayleigh/Mie wind mean absolute bias"

*line 202: typo "positively"*

> changed accordingly:  positive --> "positively"

*line 227: typo "shows" and "but"*

> changed accordingly:  show --> "shows", bu --> "but"

*line 304: typo "estimate"*

> changed accordingly:  estimates --> "estimate"

*line 359: typo "do not"*

> changed accordingly:  don't --> "do not"

*line 398: typo "is not" - Figure 5: Please add the month name into the panels of the figure*

> no changes, as we think it is sufficient that the month names are mentioned in the caption of the Figure

*line 414: typo "state"*

> changed accordingly: states --> "state"

---

## Author Comment (AC2) · 17 Jan 2021

**Response to Referee Comment (RC2) on**

**Validation of Aeolus winds using radiosonde observations and NWP model equivalents**

*https://doi.org/10.5194/amt-2020-404*

We are grateful for the referee's careful reading and the detailed and insightful discussion on our manuscript. The responses to the individual comments and the corresponding changes in the manuscript are presented in the following.

**General Comment:**

*The manuscript is well structured and addresses an important topic of the new observation system which is of significant importance for the NWP. It is found as an important contribution to the Aeolus special edition and is well in the scope of the scientific journal. The methods applied are well designed. It is appreciated that a reference data from two independent models are provided, which allows to address the issue of the model related bias as current bias correction methods are still dependent on the model itself. The estimate of the various error sources is explained, however, the section on the representativeness error is not very clear (please see next paragraph for details). Below I list several minor issues that should be resolved. Some suggestions are as well provided*

**Specific Comments:**

**Comment #2.1:**

*Abstract is overall well understood. However, I have some questions: The bias correction applied in April 2020 is the M1 temperature correction, however, in the abstract the longitude-latitude related bias correction is described. I found this confusing when reading the abstract and I suggest modifying sentence on line 20 such that it is clear that the operational bias correction at ECMWF is not the one studied in the manuscript.*

Response to Comment #2.1:

We agree, that the information in the abstract about the operational bias correction after the brief description of the longitude-latitude bias components investigated in the study can be confusing for the reader. To make clear that the L2B processor bias correction applied in April 2020 is different to the correction approach shown in the following paper the detail, that the operational bias correction is based on the telescope temperature is added:

**Since 20 April 2020 a telescope temperature-based bias correction scheme has been applied operationally in the L2B processor, developed by the Aeolus Data Innovation and Science Cluster (DISC).**
* * *
**Comment #2.2:**

*What does expression "analytical" (Line 49) represent? I don't see it necessary and I suggest removing it.*

Response to Comment #2.2:

The message of the whole paragraph is, that validation studies, that are necessary to acquire knowledge of statistical errors, are based on a systematic investigation of comparisons with collocated observations (reference data sets). The expression analytical means, that all decisive factors and components are to be taken into account for the comparisons. To avoid misunderstandings the expression "analytical" is replaced:

**For this purpose, uncertainty assessment and validation through extensive comparisons with reference data is an essential requirement to assimilate these novel observations in NWP models and fully exploit the provided wind information.**
* * *
*Comment #2.3:*

*I believe in methodology section it should be mentioned what is the time span of the validation (i.e. validation period). It is mentioned in terms of baseline but not in terms of date-time, which is important for readers not directly related with the Aeolus mission.*

Response to Comment #2.3:

The following sentence in the section Data and Method (line 98ff.) already describes the validation time period of this study in terms of date-time (month and year), which includes three mission phases:

***Furthermore, the instrument performance varied over the time period assessed in this study, which includes the missions Commissioning Phase (CP) from launch until the end of January 2019, the late Flight Model A (FM-A) laser period until mid of June 2019, and the FM-B laser period until the end of December 2019.***

Furthermore, the time periods of the different baselines are shown in Figure 1.
* * *
*Comment #2.4:*

*A word on quality control. This is first mentioned in (Line 106). But it is confusing because it is not clear what all is part of quality control? Is it defined just by L2B HLOS error estimate or anything else? There were problems with hot pixels in FM-A period, which has been handled with specific QC (for example). I suggest adding few sentences about this in case more complex QC has been used (in methodology).*

Response to Comment #2.4:

For the Aeolus data, all parts of quality control are described in the paragraph in the end of section 2.1.: To make this more clear the word quality control criteria is mentioned in an additional sentence before this part:

**For the validation, the following quality control criteria are applied. Only valid Rayleigh clear and Mie cloudy winds (from now on referred to as Rayleigh and Mie) between 800 and 80 hPa are used. A distinction is made between the ascending...**

Further, to point out how many data pass the quality monitoring used in this study, the following information is added:

**Based on a compromise between the quality of the data set and the number of observations that pass the quality control, Rayleigh winds with an estimated error greater than 6 m s⁻¹ and Mie winds with an estimated error greater than 4 m s⁻¹ are excluded. Thus, on average over the validation period about 70 % of the Rayleigh and 76 % of the Mie winds are available for the analysis.**

Because there is a preprint paper about the hot pixel issue since November 2020, we added the following sentence and reference:

**On June 14, 2019 a correction scheme for dark current signal anomalies of single pixels (hot pixels) on the Accumulation-Charge-Coupled Devices (ACCDs) of the Aeolus detectors has been implemented into the Aeolus operational processor chain (Weiler et al., 2020). All measurements before June 14, 2019 affected by hot pixels are excluded from the validation statistic.**

**Weiler, F., Kanitz, T., Wernham, D., Rennie, M., Huber, D., Schillinger, M., Saint-Pe, O., Bell, R., Parrinello, T., and Reitebuch, O.: Characterization of dark current signal measurements of the ACCDs used on-board the Aeolus satellite, Atmospheric Measurement Techniques Discussions, 1–39, https://doi.org/10.5194/amt-2020-458, https://amt.copernicus.org/preprints/amt-2020-458/, 2020.**
* * *
*Comment #2.5:*

*The sentence given in lines (145-147) is not clear. What is the meaning of ". . . 10 deg. Latitude x 10 deg. Longitude and limited periods of 7 day"? I am not sure how to interpret this information. Could this be better explained?*

Response to Comment #2.5:

We agree, that the description of how the global statistic is calculated should be explained and motivated in more detailed. Therefore, we added the following part at the end of Section 2.3.:

**To put the regional validation results in a global context, a global statistic with the ECMWF O-B values is calculated, additionally. For this, a similar approach of limited regions and limited time periods is chosen. O-B statistics are calculated for regions of 10° latitude x 10° longitude and over periods of seven days before they are averaged for the whole globe, to reduce the influence of horizontal and temporal fluctuations of systematic errors on the random errors.**
* * *
*Comment #2.6*

*In the collocation methodology of radiosondes. What is the reason for choosing the 500 m in vertical (Line 123)? Is this some sort of trade-off to have enough large sample? The HLOS range-bins are thinner near the ground especially for Mie.*

Response to Comment #2.6:

In the mid and in the upper troposphere/lower stratosphere the range bins are up to 2000 m. The winds below 800 hPa are excluded in this validation study due to noisy signal and poor data quality (especially before changes in Mie resolution). The 500 m collocation criteria is not based on a specific condition, but it seems to be a meaningful choice to ensure a good balance between sample size and accuracy.
* * *
*Comment #2.7*

*For the model description given in Section 2.3 I suggest adding the information on horizontal resolution of both models (like it is provided for COSMO and ICON LES in the text below). I think this is important for the understanding the representativeness error of HLOS winds when compared to various models.*

Response to Comment #2.7:

Following the reviewer's suggestion, we added the information about the horizontal resolution of the global model ICON from DWD and the ECMWF IFS model:

**The first guess forecast of the deterministic ICON with approximately 13 km horizontal resolution is used to calculate the observation first guess departures (O-B). In contrast to the ECMWF data assimilation system (4D-Var) with a resolution of approximately 9 km, the observations are not used at their actual time, but all observations within an observation window (±1.5 hr around the analysis time) are assumed to be valid at the analysis time.**
* * *
*Comment #2.8*

*The methodology on the estimation of representativeness error is not very clear (section 2.4). The same for results give in Section 3.2.1. It is not clear how the estimation of the various representativeness errors is actually performed (for the purpose of reproducibility of data presented in manuscript). How exactly are two high resolution models used to estimate the representative error of HLOS if first L2B HLOS is compared to global model (Eq. 6b) and second to radiosondes (Eq. 6a)? What is a definition of point and line measurement in the scope of estimation of representativeness error (Line 250)? I don't understand how values are computed (Line 254, 255, 264). Values reported in Lines 254,255 are found as sigma_b in Table 2. Please clarify. I suggest providing more detailed explanation of the methodology used (in section 3.2.1) in section 2.4?*

Response to Comment #2.8:

We agree that the estimation of the representativeness error can be explained more clearly. Therefore, the explanation in the methodology part is updated to provide more information about the different error components contributing to the representativeness error:

**2.4 Representativeness errors for the Aeolus wind validation**

[revised manuscript text omitted]

**the Mie winds with 90 km horizontal resolution and 2.66 m s⁻¹ for the Mie winds with 10 km horizontal resolution.**
* * *
*Comment #2.9*

*In line 183 (section 3.1 in general) and Table 1 I found some confusing information about the term "mean absolute difference" (and mean absolute bias). What exactly is computed here? Is this an average of absolute values of BIAS shown in Figure 1. If so, in Table 1 |BIAS| is confusing?*

Response to Comment #2.9:

Table 1 shows the average of the absolute values of Figure 1.  To be exact, these values are differences, but like mentioned in Section 2.5. ***(Given that the model bias for long validation periods and large scales is usually small in comparison to that of Aeolus observations, the mean difference between the Aeolus observations and the reference data can be referred to as bias.)*** the differences between the Aeolus observations and reference data sets can be referred to as bias. To avoid confusion the absolute value bars are removed in the table. The caption already gives the information that the mean absolute bias estimates are shown.
* * *
*Comment #2.10*

*At the end of Line 195: " . . . of limited areas (10 deg. Latitude . . .". This is related to the comment given before. I don't understand what is meant by this. How are these limited areas used for computation of global statistics? (See please the same comment above).*

Response to Comment #2.10:

See Response #2.5
* * *
*Comment #2.11*

*Line 197. I would not agree for Mie descending. But I would say that the values for global statistics are in the range of values of all three (radiosondes, ECMWF and DWD) local statistics. Which, on the other hand, does not hold for Rayleigh-clear statistics.*

Response to Comment #2.11:

We agree, that the estimated global mean absolute bias for Mie is in the range of all three local statistics. Due to the fact, that all three local statistics include only values around the radiosonde collocations (see Section 2.3. : ***To ensure comparable data sets for the radiosonde and the ECMWF and DWD model validation of the Aeolus winds, only the nearest O-B value per radiosonde collocation is used for the model validation statistics.)*** , the expression 'validation statistics around the radiosonde collocations' includes all three local statistics. To avoid any misunderstandings, we slightly modified the sentence:

**For the Mie winds, the global statistic shows values in the range of the three local validation statistics around the radiosonde collocations.**
* * *
*Comment #2.12*

*Line 218. Which bias correction update? I suggest being specific and to mention that this is the one that is validated in the following sections (the lon-lat dependent one).*

Response to Comment #2.12:

The L2B processor bias correction update in December 2019, which is mentioned here, is not the one validated in the following. It was a manual bias correction of + 4 m/s in the Rayleigh wind product to compensate for a global average bias drift, which is due to another effect - a changing Rayleigh internal reference response. To be more precise, this information is added:

**This is caused by a manual L2B processor bias correction of + 4 m s$^{-1}$ in the Rayleigh wind product to compensate for a global average bias drift.**
* * *
*Comment #2.13*

*In Table 2 the sigma_b for Mie on 90 km is about 0.5 m/s but only 0.12 m/s when on 10 km. What is a scientific reason for this? Because no matter the Mie accumulation the IFS HLOS is always an interpolation to the L2B HLOS center of gravity location. So is this the result of QC used or maybe better L2B classification on clear cloudy for smaller accumulations?*

Response to Comment #2.13:

Due to the smaller horizontal resolution there are a lot more Mie HLOS wind observations which reduces the error, especially in cloudy sky condition. To address the differences in $\sigma_b$ (Mie) the following passage is added in section 3.2.2.:

**As NWP models in general tend to exhibit higher uncertainty in cloudy than in clear sky areas, $\sigma_b$ is larger for the Mie winds with 90 km horizontal resolution. After the decrease of the horizontal integration length of the Mie wind measurements to approximately 10 km in the L2B product, the number of Mie wind observations increased, leading to a reduction in model error.**
* * *
*Comment #2.14*

*Line 295: Sentence started with "Since the representativeness . . ." is not clear. What does it mean? What is the range of estimated uncertainties?*

Response to Comment #2.14:

We agree that this part should be formulated more exactly. It has been modified as follows:

**Because the estimation of the representativeness error is based on averaged values of analyses which only cover the area around Germany at certain time periods, the values are affected by small uncertainty factors. As the model error estimates are also associated with uncertainty, it is assumed that the discrepancies between the radiosonde and model validation are due to uncertainties in the calculation of the different error sources.**
* * *
*Comment #2.15*

*Line 337: It is confusing how the bias correction is applied (Eq. 7). "i=1 being the current day" suggests that at the day of applying the bias correction the O-B for that same day are as well included. Isn't that the bias correction factor for a particular day is provided from O-B of previous days?*

Response to Comment #2.15:

This wrong information is changed: **i = 0 being the the current day**
* * *
*Comment #2.16*

*Figure 6: the pattern looks very random. Are these differences statistically significant (especially for ascending orbit)?*

Response to Comment #2.16:

Figure 6 shows the absolute differences between the ECMWF IFS and the ICON model O-B values, exemplary for May 2019. With this plot we want to demonstrate that the differences are indeed not statistically significant (as the reviewer has noted) and thus justify that the O-B statistics as function of latitude and longitude (Figure 5) are only presented for the ECMWF first guess departures.
* * *
*Comment #2.17*

*How does the computed estimate of the Aeolus instrument error compare with the L2B estimated HLOS instrument error? By the values reported in the manuscript this seems quite well in range, but I miss this comparison. It would be of interest to see how good is the L2B error estimation algorithm. Has this been studied?*

Response to Comment #2.17:

It turned out, that the L2Bp estimated error has to be treated carefully due to a bug until the baseline change in April 2020.  Basically, it overestimated the error in daylight conditions (i.e., large UV solar background noise). Therefore, if looking at the NH in summer (late FM-A) the estimated errors are all too large. However, this issue is not fully understood yet and there are ongoing studies regarding the L2Bp estimated error. Consequently, no comparisons between the observation instrument noise estimate and the estimated instrumental error are made in this paper, which would be also beyond the original scope of the paper. It is agreed that a thorough investigation of the L2B error estimates would deserve a separate study
* * *
*Comment #2.18*

*In addition, for Conclusions. The bias correction method used is essentially a temporal and spatial smoothing. This is as well a source of bias residuals shown in Figure 4 and 6, as bias correction method is not able to react on fast changes in systematic errors that happens along the Aeolus orbit.*

Response to Comment #2.18:

We appreciate the reviewer's comment about the bias correction method. We agree that the temporal and spatial smoothing of the correction approach should be mentioned in the text. We added the following lines in  the summary and discussion part:

**As the bias correction approach is essentially a temporal and spatial smoothing, it is suggested that fast changes in systematic errors are one source of the bias residuals.**

***Further modification:***

The following reference has been added to the introduction part as example for already published Aeolus wind validation studies:

**Baars, H., Herzog, A., Heese, B., Ohneiser, K., Hanbuch, K., Hofer, J., Yin, Z., Engelmann, R., and Wandinger, U.: Validation of Aeolus wind products above the Atlantic Ocean, Atmospheric Measurement Techniques Discussions, 2020, 1–27, https://doi.org/10.5194/amt-2020-198, 2020.**

**Technical stuff**

*A general note on Figures. I suggest to use the same framework on all images. This is by using labels (a,b . . .). At the moment this is not true everywhere and on some images (Right, Left, . . .) is used instead. In addition, on some Figure there is no units, this should be corrected.*

*Line 253: Please use the uniform Figure labels (i.e. a,b,. . .)*

*Figure 2: Please use labels a,b,. . .*

*Figure 5: missing labels*

*Figure 6: missing labels*

> changed accordingly: labels (a), (b), (c) are added to Figure 2, units are added to Figure 5 and 6

*Line 8: instead of "comparisons" it would be maybe better to use "the validation"*

> no changes , as we think the current wording is more appropriate

*Line 11: If the "independent reference data sets" are the two models, I suggest here to be specific and change this with ". . . between the two model data sets"*

> changed accordingly: "between the three independent reference data sets"

*Line 11: I suggest to switch "representativeness" into "representative of Aeolus winds"*

> changed accordingly: "Due to the greater representativenesses errors associated with the comparisons using radiosonde observations, the random differences are larger for the validation with radiosondes compared to the model equivalent statistics."

*Line 13: Sentence (To achieve . . .) is not well understood. I suggest turning it around "Besides the . . . the Aeolus instrument error is estimated . . ." or similar*

> no changes, as we think the current wording is more appropriate

*Line 16: The expression "depend on" should probably be changed to "vary with"*

> changed accordingly: depend on --> "vary with"

*Line 25: Better be specific. I suggest to switch "Earth's wind" to "atmospheric wind"*

> changed accordingly: Earth's wind fields --> "atmospheric wind fields"

*Line 26: The sentence "Within seven days, . . ." is not completely correct. The orbit does not cover the globe as the footprint is in scale of meters and orbits are separated by 1000 km or more. But the repeat cycle of the orbit is about 7 days. I suggest to rewrite this*

> changed accordingly: *Within seven days --> "Within about seven days"*

*Line 27: Could remove the "only one large instrument", it is not necessary*

> no changes, as we think the current expression is more appropriate

*Line 31: First two sentences could be switched. It reads better*

> no changes, as we think the current order of the sentences reeds better

*Line 36: Could remove "on" and "parts"*

> no changes, as we think the current wording is more appropriate

*Line 42: At the end of the sentence ". . . geostrophic balance" I suggest to put some reference. For example, https://doi.org/10.1175/BAMS-86-1-73*

> the following reference is added:  Stoffelen, A., Pailleux, J., Källén, E., Vaughan, M., Isaksen, L., Flamant, P., Wergen, W., Andersson, E., Schyberg, H., Culoma, A., Meynart, R., Endemann, M., and Ingmann, P.: The Atmospheric Dynamics Mission for Global Wind Field Measurements, Bulletin of the AmericanMeteorological Society, 86, 73–88, 2005

*Line 45,46: I suggest to add few additional valuable references (https://doi.org/10.1256/qj.05.83, https://doi.org/10.1002/qj.43, https://doi.org/10.1002/qj.2430).*

> *no changes, as we think the five references are sufficient here.*

*Line 67: I suggest to put a new paragraph just before "The text is struct. . ."*

> a new paragraph is insert

 *Line 71: "representativeness error of comparisons", this is very confusing. Could the expression "comparisons" be removed here?*

> changed accordingly: "The derivation of error estimates for the Aeolus instrumental error includes the determination of representativeness errors which is based on analysis data from..."

*Line 80: In reference "ECMWF: . . ." the "ECMWF:" should be removed.*

> changed accordingly: ECMWF is removed in the reference

*Line 86: "The resulting HLOS wind" should be modified by "The resulting HLOS wind observation therefore represents a horizontal average over . . ."*

> changed accordingly: "The resulting HLOS wind observation therefore represents a horizontal average over 86.4 km."

*Line 87: The reference "Matic et al" should be changed to "Šavli et al"*

> reference is modified accordingly: Matic et al. --> "Šavli et al."

*Line 87 (end): "the" should be removed.*

> changed accordingly: "In addition to HLOS observations, ..."

*Line 89: ". . ., to avoid systematic errors. . ." should be changed to " . . . to reduce the HLOS systematic error . . ." (for example). This is because there are other sources of systematic error that exist. Line 93: I suggest to change the ". . . types" to "... resulting in four wind products (i.e. Rayleigh-clear, Rayleigh-cloudy, Mie-clear and Mie-cloudy)." to be more specific.*

> changed accordingly: "Furthermore, to reduce systematic errors, ..."
> changed accordingly: "Wind retrievals are performed for both channels resulting in four observation products (Rayleigh-clear, Rayleigh-cloudy, Mie-clear and Mie-cloudy)."

*Line 95: Again the reference "EMCWF: . . ." should be corrected by removing "ECMWF:"*

> changed accordingly: ECMWF is removed in the reference

*Line 115 (and 116): I suggest to rewrite a bit into "...is taken into account by splitting data into groups of 15 min."*

> changed accordingly: "... by splitting data into groups of 15 min ."

*Line 116 (by the end): Please correct "contains" into "contain".*

> changed accordingly: contains --> "contain"

*Line 118: Please correct "the ascent and the ascent" into " the ascent and the descent".*

> changed accordingly: "Due to the radiosonde drift during the sounding and the ascent time, additional errors arise."

*Line 156: I suggest to change the heading title into: "statistical measures", "statistical metrics", "validation metrics", or something similar. "Statistical characteristics" is not valid for description of statistical metrics that have been used. For Eq. 2 I suggest to add that index "i" represents time. Eq 3-4 seem incomplete. In results bias is provided $_{for}$ radiosonde, and 2 models. I suggest to explain here that different biases and STD are provided for different reference data. This should be seen by equations defined here. At the moment it seems that Eq 6 for STD is always computed taking BIAS from model (even if STD is computed for radiosondes). I believe this is not what has been used. Correct me please if I am wrong. Eq 5 is as well confusing. I suggest to replace this with the definition for MAD (so removing the "scaled"), which is a very basic measure. And in the text it should be mentioned that scaled MAD is used instead which is defined as 1.4826\*MAD because it provides a measure similar to STD for normal samples.*

> changed accordingly: Statistical characteristics --> "Statistical metrics"
> the indices used in the equations are explained by adding the following phrase: "...,where i represents the time step and N is the number of compared data points."
> to avoid confusion, we defined $v_{diff(HLOS)}$ in Eq 2 and 3 to represent the difference between the Aeolus L2B HLOS wind and the HLOS wind value of the different reference data sets. $v_{diff(HLOS)}$ is also used in the equation for STD and the scaled MAD to make them easier to understand. The following sentence is added after Eq 4 and 5: "For quantifying random deviations, the standard deviation (Eq4) as well as the scaled median absolute deviation (MAD)(Eq 5) is determined for the three reference data sets."

*Title of heading in section 3 is confusing. It suggest shorten it. Maybe something like "Aeolus HLOS error time series characteristics".*

> we agree, that the title is somehow too long. It is changed accordingly: "Validation results - time series characteristics and error estimation of Aeolus HLOS wind comparisons". The time series (Section 3.1) show the results of the comparisons which are, to be correct, differences and not errors. Subsequently (Section 3.2), estimated error values are determined.

*Line 199: Please modify ". . . wind bias, . . ." to "...wind bias and random error, . . ." Line 204: "modes" is probably "mode"?*

> unchanged, because the following part focuses only on the systematic differences. Random differences are discussed before
> changed accordingly: standby modus --> "stand-by mode"

*Line 227: Please correct "bu" to "but"*

> changed accordingly: bu --> "but"

*Line 237: Please correct "differences" to "difference"*

> changed accordingly: differences --> "difference"

*Line 253: Please use the uniform Figure labels (i.e. a,b,. . .)*

*Figure 2: Please use labels a,b,. . .*

*Table 2: In caption "...(Representativeness errors . . ." should be corrected to " ...(representativeness errors ...)"*

> changed accordingly: (*Representativeness errors...*) --> (*representativeness errors...*)

*Line 329: "Thus, it is tested . . ." should be correct to "Thus, it is first tested . . .."*

> changed accordingly: T*hus, it is tested* --> "Thus, it is first tested..."

*Line 401: I think this is not needed as this is already given in the last paragraph of Introduction.*

> no changes, as we think with this information it reeds better

*Line 406 (at the end): Please correct "as" to "as a"*

> no changes , as we think the current wording is more appropriate

*Line 480, 483 and 485: Please check if this kind of referencing is correct.*

> changed accordingly: ECMWF is removed in the reference (Rennie, M., Tan, D. G. H., Andersson, E., Poli, P., Dabas, A., de Kloe, J., Marseille, G.-J., and Stoffelen, A.: Aeolus Level-2B Algorithm Theoretical Basis Document (Mathematical Description of the Aeolus Level-2B Processor), AE-TN-ECMWF-L2BP.0023, V.3.21, 2020.).

> The ESA ADM-Aeolus Science Report and the ESA ADM-Aeolus Mission Requirements Documents are referenced in the same way for example in Lux et al. (2020) and Witschas et al. (2020)

*Line 502: Please correct "Matic. S" to "Šavli. M" and "Zagar. N" to "Žagar. N"*

> reference is modified

*Line 536: Please correct "Zagar. N" to "Žagar. N"*

> reference is modified

---

## Referee Report (RR1)

**Review of manuscript "Validation of Aeolus winds using radiosonde observations and NWP**

**model equivalents" by Anne Martin et al**

**General comments**

Authors provided very good update on the manuscript taking into account previous comments and suggestions. The description of the representativeness error is very clear after this update. With exception of two minor technical comments, added below, there is no need for further corrections.

**Techical corrections:**

Line 274: windmodel → wind model

Comment on response of comment #2.8: Representativeness error in table 2. In Line 281-282 it is reported that the $\sigma_{r\_model}$ is 0.5 m/s, 0.52 m/s and 0.12 m/s. However in Table 2, these same values appear as $\sigma_b$, which is rather confusing. Are values in table 2 correct? Please ignore the comment if it is correct.